# TORQUE-AWARE MOMENTUM

## ABSTRACT

Efficiently exploring complex loss landscapes is key to the performance of deep neural networks. While momentum-based optimizers are widely used in state-of-the-art setups, classical momentum can still struggle with large, misaligned gradients, leading to oscillations. To address this, we propose Torque-Aware Momentum (TAM), which introduces a damping factor based on the angle between the new gradients and previous momentum, stabilizing the update direction during training. Empirical results show that TAM, which can be combined with both SGD and Adam, enhances exploration, handles distribution shifts more effectively, and improves generalization performance across various tasks, including image classification and large language model fine-tuning, when compared to classical momentum-based optimizers.

## 1 INTRODUCTION

Despite the wide range of optimization methods available in the literature, stochastic gradient descent (SGD), typically augmented with momentum (Kingma & Ba, 2015; Nesterov, 1983; Qian, 1999), remains the go-to approach for practitioners. Momentum accelerates convergence, particularly in the presence of high curvature (Cutkosky & Mehta, 2020b), small but consistent gradients, or noisy gradients. It also helps the optimizer navigate the loss landscape and escape local minima or saddle points by maintaining consistent updates directions (Jin et al., 2018).

While SGD with momentum (SGDM) has shown remarkable success in various scenarios, particularly in computer vision (Sutskever et al., 2013), it remains vulnerable to the adverse effects of large, misaligned gradients (Zhang et al., 2019). These gradients often stem from noisy data or abrupt changes in loss landscape curvature, especially in narrow basins where gradients frequently shift direction (Ortiz-Jiménez et al., 2022). This can lead to oscillations, making it harder for the optimizer to escape sharp minima (Fu et al., 2023).

In this work, we propose that minimizing the influence of misaligned gradients during momentum updates can preserve valuable information and improve the exploration capabilities of momentum-based methods. To enable more consistent exploration of the loss landscape, particularly in noisy settings, we introduce a new approach that modifies the standard momentum update by incorporating a damping factor, inspired by the damping effect in mechanical systems (Fritzen, 1986).

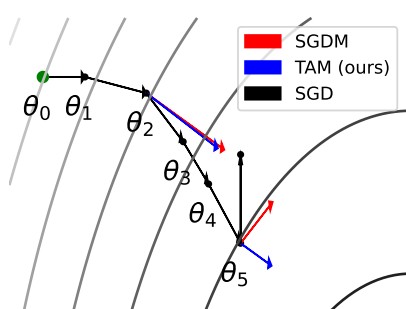

Figure 1: Comparing momentum updates obtained using SGDM and TAM for a given SGD trajectory. While TAM results in more stable directions pointing to a lower loss basin, SGDM has higher magnitude updates susceptible to misaligned gradients.

In this analogy, momentum represents velocity in linear dynamics, and the gradient represents the applied force. The damping term we introduce depends on the angle between the gradient and momentum, acting as anisotropic friction (Tramsen et al., 2018). This term modulates the influence of misaligned (or 'torqued') gradients, much like damping reduces torque in rotational systems. Drawing from this physical analogy, we name our method Torque-Aware Momentum (TAM).

Figure 1 illustrates how TAM (blue) modifies the momentum update in terms of both magnitude and direction compared to SGDM (red) along an SGD trajectory (black). At $\theta_2$, where the gradient aligns with the previous momentum, both SGDM and TAM incorporate the new gradients similarly, propelling the parameters forward. However, at $\theta_5$, where a misaligned (torqued) gradient emerges, SGDM's update direction shifts abruptly due to the conflicting gradient. In contrast, TAM maintains stability by preserving the previous momentum direction, allowing for continued exploration without discarding past information.

Our empirical analysis shows that this consistent exploration early in training helps discover more generalizable basins in the loss landscape. Our key contributions are as follows:

- We propose Torque-Aware Momentum (TAM), a new method that mitigates the impact of torqued gradients while enhancing exploration in momentum-based optimizers (Section 3).

- We illustrate the performance of TAM and its adaptive variant, AdaTAM, with experiments on image classification tasks using CIFAR10, CIFAR100, and ImageNet (Section 4.1) as well as fine-tuning different large language models (Section 4.2).

- We demonstrate additional benefits of TAM, specifically its increased robustness to distribution shifts in online learning setups (Section 4.3) and its effectiveness as a warm-up phase to enhance exploration in the early stages of training (Section 4.4).

## 2 RELATED WORK

Momentum-based methods have been widely studied for their ability to improve convergence speed and exploration of the loss landscape. For instance, Xing et al. (2018) showed that as mini-batch gradients aligns with the top eigenvectors of the Hessian, SGD's exploration slows due to oscillatory behaviour, particularly at larger batch sizes. Similarly, Fu et al. (2023) showed that SGDM accelerates convergence by deferring this oscillation, referred to as *abrupt sharpening*, where gradients and the Hessian suddenly align, making SGDM more effective for larger learning rates.

Several momentum variants aim to improve generalization by utilizing the curvature of the loss surface (Gilmer et al., 2021; Foret et al., 2021; Yao et al., 2021; Tran & Cutkosky, 2022; Kaddour et al., 2022).

Popular optimizers like Adam (Kingma & Ba, 2015) combine adaptive learning rate with momentum for faster convergence, while Ziyin et al. (2020) proposed leveraging parameter updates, rather than gradients, to compute momentum. However, while these methods improve convergence speed, they do not specifically address the challenge of torqued gradients on noisy loss surfaces.

Lucas et al. (2018) introduced AggMo, an optimizer combining multiple momentum vectors with different decay rates, but requires storing multiple copies of model states (Cutkosky & Mehta, 2020a; Xie et al., 2021), unlike our method TAM, which maintains the same memory footprint as SGDM. Closest to our work, S.K. Roy & Chaudhuri (2021) tackle gradient misalignment by considering angles between consecutive gradients. However, we argue that focusing on the angle between momentum and gradients is more critical for stability, as demonstrated by our comparisons with their method, AngularGrad (seeSection 4).

## 3 METHODOLOGY

**Background: SGDM** Momentum was first introduced to accelerate convergence in SGD (Polyak, 1964; Qian, 1999). Given a loss function $L_D(\theta)$ and its gradients $g_t = \nabla_{\theta_t} L_D(\theta_t)$ at time $t$, the momentum and parameter updates are:

$$m_t = \beta m_{t-1} + g_t; \quad \theta_{t+1} = \theta_t - \eta m_t \tag{1}$$

where $\beta$ is the momentum coefficient and $\eta$ is the learning rate. The momentum accumulates past gradients, smoothing out noise and providing more weight to recent gradients. This helps accelerate convergence by allowing the optimizer to maintain a consistent update direction, even in the presence of noisy gradients or small gradients from the mini-batches (Sutskever et al., 2013).

**Torque-Aware Momentum (TAM)**  TAM modifies the momentum update in Eq. 1 to regulate the impact of new gradients. To handle the noisy nature of loss surfaces, we introduce a damping factor that adjusts the influence of gradients based on their directional alignment with the previous momentum. This acts like anisotropic friction (Tramsen et al., 2018), reducing the effect of torqued gradients, similar to how damping reduces torque in rotational systems.

To increase robustness against misaligned gradients and encourage exploration of dominant gradient directions, we define the correlation $S_t$ between the previous momentum direction and the current gradient as the cosine similarity:

$$S_t = \frac{m_{t-1} \cdot g_t}{||m_{t-1}|| ||g_t||} . \qquad (2)$$

We apply smoothing to $S_t$ with a decay rate $\gamma$ to account for stochasticity:

$$\hat{s}_t = \gamma \hat{s}_{t-1} + (1 - \gamma) S_t . \qquad (3)$$

Next, we normalize the smoothed correlation $\hat{s}_t$ to the range $[0, 1]$ and introduce a small constant $\epsilon$ to ensure that new gradients still exert a small influence even when the momentum magnitude diminishes. We prioritize momentum update aligned with previous directions to reduce the influence of large opposing gradients:

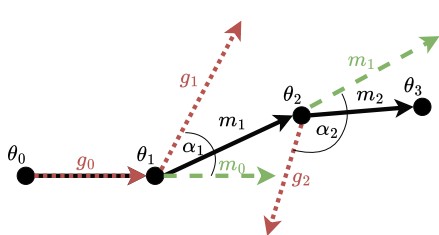

Figure 2: **TAM controls update magnitude (red) based on the alignment between momentum and new gradients.** The angle ($\alpha_1$, $\alpha_2$) between previous momentum (green) and new gradients (white) determines the magnitude of the update (red). When $g_1$ aligns well with $m_0$, the resulting momentum $m_1$ has a higher magnitude. In contrast, when the misalignment between $g_2$ and $m_1$ results in a smaller magnitude $m_2$.

$$d_t = \frac{1 + \hat{s}_t}{2} \;;\quad m_t = \beta m_{t-1} + (\epsilon + d_t) g_t . \qquad (4)$$

Though TAM introduces the hyper-parameters $\gamma$ and $\epsilon$, they are fixed by default at $0.9$ and $1e - 8$, respectively, requiring no additional tuning. Figure 2 illustrates TAM's behaviour: when the alignment $\alpha_1$ is stronger (smaller $\alpha_1$), the gradient $g_1$ amplifies the momentum $m_1$. Conversely, when $\alpha_2$ is larger, the gradient $g_2$ has less influence, resulting in a smaller momentum $m_2$. The pseudo-code of TAM is given in Algorithm 1.

**Learning Rate Transfer**  Here we describe a simple heuristics to transfer a tuned learning rate from SGDM to TAM. We can do so by comparing effective learning rates, as derived in (Fu et al., 2023). For SGDM, the idea is that momentum changes the update magnitude in a way that can be approximated as $t$ gets large as

$$m_t = \sum_{s=1}^{t} \beta^{t-s} g_s \approx \frac{1 - \beta^t}{1 - \beta} g_t \to \frac{1}{1 - \beta} g_t$$

This suggests that the SGDM updates (1) with learning rate $\eta$ have the same magnitude as the updates of SGD with effective learning rate $\eta_{\text{SGDM}}^{\text{eff}} = \frac{1}{1-\beta} \eta$. Similarly, we derive the effective learning rate for TAM based on the update rule (4) with $||\epsilon|| \ll 1$. Assuming that, as $t$ in-

---

**Algorithm 1** TAM update

**Require:** Initial parameters $\theta_0$, momentum $m_0$, learning rate $\eta$, momentum coefficient $\beta$, smoothing decay rate $\gamma$, $\epsilon$, # of iterations $T$.
$\hat{s}_0 = 0$
**for** $t = 1, 2, \ldots, T$ **do**
  Sample mini-batch $b_t$ from data $\mathcal{D}$
  Compute gradients $g_t = \nabla_{\theta_t} L_{b_t}(\theta_t)$
  $S_t = m_{t-1} \cdot g_t / ||m_{t-1}|| ||g_t||$ (Eq. 2)
  $\hat{s}_t = \gamma \hat{s}_{t-1} + (1 - \gamma) S_t$ (Eq. 3)
  $d_t = (1 + \hat{s}_t)/2$
  $m_t = \beta m_{t-1} + (\epsilon + d_t) g_t$ (Eq. 4)
  $\theta_t = \theta_{t-1} - \eta m_t$
**end for**
**return** $\theta_T$

---

creases, the cosine similarity $\hat{s}_t$ stabilizes to a constant value $s^*$, TAM's effective learning rate becomes:

$$\eta_{\text{TAM}}^{\text{eff}} \approx \frac{1 + s^*}{2(1 - \beta)} \eta \qquad (5)$$

Under this assumption, a tuned learning rate $\eta_{\text{SGDM}}^*$ for SGDM can be transferred to an optimal learning rate $\eta_{\text{TAM}}^*$ for TAM by equating the corresponding effective learning rate. Solving for $\eta_{\text{TAM}}^*$

yields:

$$\eta^*_{\text{TAM}} = \frac{2(1 - \beta_{\text{TAM}})}{(1 + s^*)(1 - \beta_{\text{SGDM}})} \eta^*_{\text{SGDM}} \cdot \tag{6}$$

In practice, we observed that $s^* \approx 0$ as $t$ increases (see Appendix A.2.1 for empirical evidence). In our experiments, we set $\beta_{\text{TAM}} = \beta_{\text{SGDM}}$, and found that $\eta^*_{\text{TAM}} = 2\eta^*_{\text{SGDM}}$ consistently yields optimal performance.

This equivalence means that in the neighborhood of optima, where $\hat{s}_t$ has stabilized, TAM inherits the well-established convergence guarantees of SGDM (Yan et al., 2018; Liu et al., 2020). The damping factor $(1 + \hat{s}_t)/2$ remains bounded, ensuring the effective learning rate stays within a controlled range throughout training. This theoretical connection to SGDM, combined with our empirical evidence of $s^*$ stabilizing to 0, ensures TAM's convergence while maintaining its enhanced exploration capabilities during early training.

**AdaTAM**   We also introduce an adaptive variant of TAM, which combines Adam (Kingma & Ba, 2015) and the TAM update in Eq. 4. The update rule for AdaTAM is thus defined as

$$m_t = \beta m_{t-1} + (\epsilon + d_t)g_t \; ; \quad v_t = \beta_2 v_{t-1} + (1 - \beta_2)g_t^2 \; ; \quad \theta_{t+1} = \theta_t - \eta \frac{m_t}{\sqrt{v_t} + c} \; , \tag{7}$$

where $\beta_2$ is the second-moment decay rate, and $c$ is a small constant (typically $1e-8$ by default). Note that AdaTAM only modifies $m_t$ and keep the updates of $v_t$ the same as in Adam.

## 4 EXPERIMENTS

In this section, we present the results of our experiments evaluating TAM across various benchmarks. First, we compare TAM and AdaTAM with baseline optimizers including SGD (with and without momentum), Adam, and AngularGrad (S.K. Roy & Chaudhuri, 2021), in terms of generalization performance on image classification datasets (subsection 4.1). We also assess AdaTAM's performance in fine-tuning Bert-based models on the MTEB datasets (subsection 4.2). Additionally, we demonstrate TAM's robustness to distribution shifts in online learning settings ( subsection 4.3) and explore its use during a warm-up phase to facilitate loss landscape exploration in the early stages of training (subsection 4.4). All results of our experiments are averaged across five seeds, with additional experimental details provided in Appendix A.1.

### 4.1 IMAGE CLASSIFICATION

**Setup.** We run experiments on CIFAR 10, CIFAR100 (Krizhevsky & Hinton, 2009), and ImageNet (Deng et al., 2009). We train ResNet18, ResNet34 architectures on CIFAR10/100 for 200 epochs and ResNet50 on ImageNet for 90 epochs. We perform a learning rate grid search with a fixed compute budget assigned to each optimizer to obtain the best setup. We choose the ranges of these grid searches to be consistent with the learning rate transfer heuristic rule in Equation 6.

**Results.** The validation accuracy for each optimizer is reported in Table 1. The results indicate that TAM and AdaTAM generally outperform their corresponding baselines across most configurations.

| | CIFAR10 | | CIFAR100 | | ImageNet |
|---|---|---|---|---|---|
| **Optimizers** | ResNet18 | ResNet34 | ResNet18 | ResNet34 | ResNet50 |
| SGD | $93.3_{\pm 0.1}$ | $93.8_{\pm 0.1}$ | $73.1_{\pm 0.3}$ | $73.6_{\pm 0.1}$ | $75.4_{\pm 0.1}$ |
| SGDM | $\underline{93.6}_{\pm 0.3}$ | $\underline{93.9}_{\pm 0.2}$ | $\underline{73.2}_{\pm 0.2}$ | $\mathbf{74.7}_{\pm 0.1}$ | $\underline{77.0}_{\pm 0.1}$ |
| TAM | $\mathbf{94.2}_{\pm 0.2}$ | $\mathbf{94.3}_{\pm 0.2}$ | $\mathbf{73.8}_{\pm 0.1}$ | $\underline{74.3}_{\pm 0.3}$ | $\mathbf{77.1}_{\pm 0.1}$ |
| Adam | $93.4_{\pm 0.1}$ | $93.6_{\pm 0.2}$ | $70.1_{\pm 0.3}$ | $71.7_{\pm 0.1}$ | $74.4_{\pm 0.5}$ |
| AngularGrad | $93.3_{\pm 0.2}$ | $93.7_{\pm 0.2}$ | $70.9_{\pm 0.2}$ | $71.2_{\pm 0.2}$ | $73.8_{\pm 0.1}$ |
| AdaTAM | $93.3_{\pm 0.3}$ | $93.3_{\pm 0.1}$ | $72.7_{\pm 0.3}$ | $72.9_{\pm 0.1}$ | $74.5_{\pm 0.1}$ |

Table 1: Comparison of TAM and AdaTAM with baseline optimizers for ResNet architectures trained on CIFAR10/100 and ImageNet with learning rate grid search.

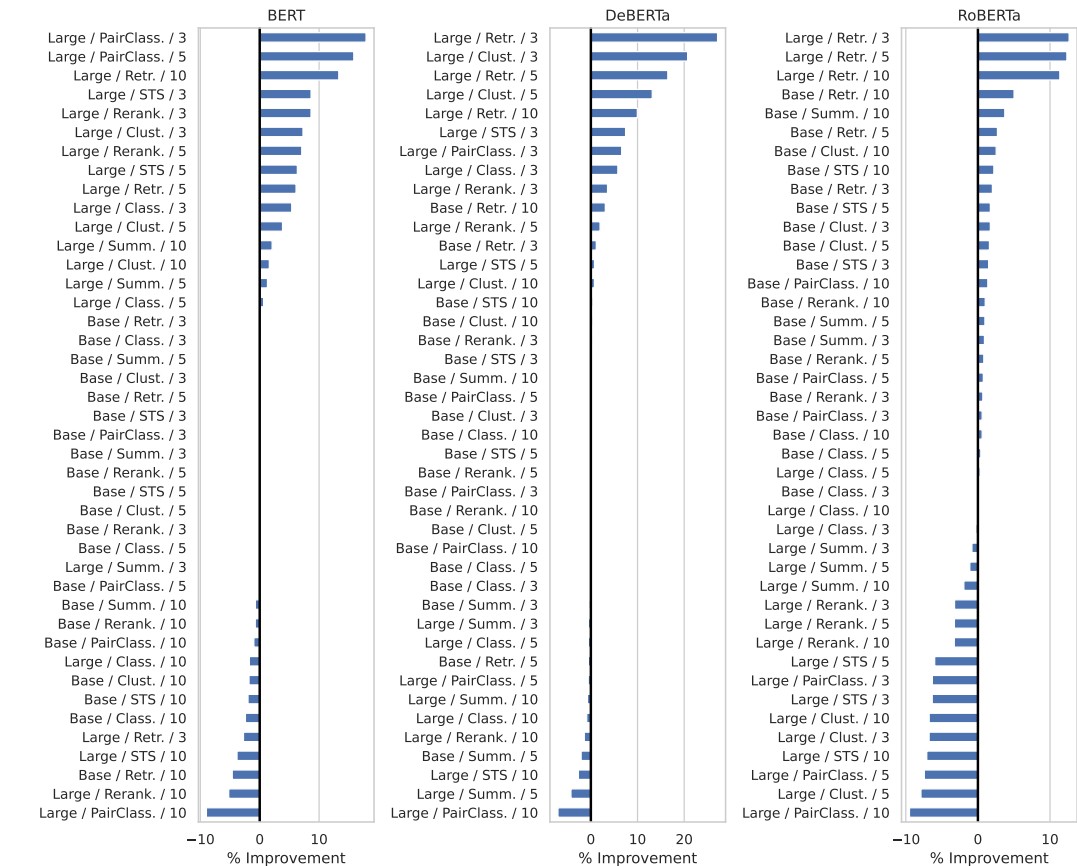

Figure 3: Percentage improvement in the average scores of AdaTAMW compared to AdamW across different MTEB task categories for three types of models: BERT (left), DeBERTa (middle) and RoBERTa (right). The the y-axis labels indicate the model size ({Base, Large}) / MTEB task category (7 in total), and the number of fine-tuning epochs ({3, 5, 10}), covering 42 configurations in total. Overall, AdaTAMW achieve similar or better performance than AdamW in at least 28 configurations across all three model types.

Among non-adaptive optimizers, the only exception is for CIFAR100 with the ResNet34 model, where TAM performs slightly below SGDM. In all other cases, TAM achieves higher accuracy. Although adaptive optimizers generally underperform compared to non-adaptive ones in these setups, we observe that AdaTAM achieves similar or even better results compared to Adam and AngularGrad, with the exception of ResNet34 on CIFAR10. Overall, while the effectiveness may vary depending on the specific model, these results indicate that TAM and AdaTAM provide consistent improvements in generalization across various models and datasets.

## 4.2 LLM FINE-TUNING

**Setup.** We compare AdaTAM with weight decay (AdaTAMW) to AdamW for fine-tuning LLMs. Specifically, we consider six pre-trained BERT-based models: BERT-base, BERT-large (Devlin, 2018), DeBERTa-base, DeBERTa-large (He et al., 2021), RoBERTa-base, and RoBERTa-large (Zhuang et al., 2021). Each model is fine-tuned on masked language modeling using the WikiText dataset (Merity et al., 2016), applying both AdaTAMW and AdamW across varying numbes of epochs. We use the open source implementation by Wolf et al. (2020). All hyperparameters, except for the learning rate, remain at their default values. A grid search was performed to identify the optimal learning rate across $\{5e-6, 1e-5, 5e-5\}$, with the best checkpoint selected based on validation perplexity. The fine-tuned models were then evaluated on the Massive Text Embedding Benchmark (MTEB), covering 7 task categories across a total of 56 datasets (Muennighoff et al., 2022).

**Results.** Figure 3 summarizes all results obtained for each type of model. Specifically, it shows the percentage improvement in the average scores of AdaTAMW compared to AdamW across the MTEB task categories for each model type. The evaluation includes a total of 42, considering two model sizes ({Base, Large}), 7 English task categories in MTEB (classification, pair classification, semantic textual similarity, information retrieval, clustering, summarization and reranking) and different fine-tuning epochs ({3, 5, 10}).

AdaTAMW shows the highest improvements over AdamW on DeBERTa models across configurations with varying numbers of epochs. In contrast, results for RoBERTa are more mixed, with the most significant improvements observed in Retrieval tasks. For BERT models, while AdaTAMW generally delivers similar or better average scores, the most notable gains occur in the 3 and 5 epoch settings. Another key observation is that AdaTAMW yields larger improvements for BERT-large and DeBERTa-large models, but it performance on RoBERTa-large is less consistent, where RoBERTa-base often outperforms it.

In addition, Figure 4 shows the percentage of times AdaTAMW performed similarly or better than AdamW.[1] Except for the RoBERTa-large and BERT-base fine-tuned on 10 epochs, AdaTAMW generally matches or exceeds AdamW's performance in most settings. Furthermore, except for DeBERTa-base, AdaTAMW achieves higher scores on more than two-thirds of the MTEB datasets for in the 3- and 5-epoch settings. Detailed results on individual MTEB datasets are reported in Appendix A.2.7.

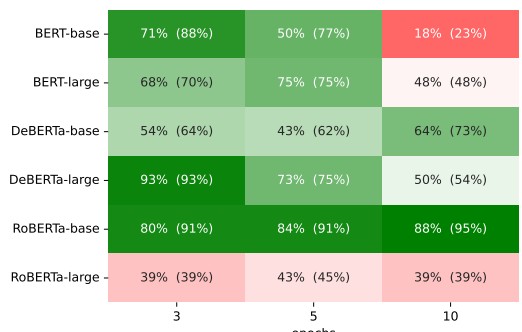

Figure 4: Percentage of times when AdaTAMW performs better (or similar/better) than AdamW on various LLMs across 56 MTEB datasets. Green indicates that AdaTAMW achieves similar or better performance, while red indicates worse performance. Except for BERT models with 10 epochs and RoBERTa-large, AdaTAMW performs similar/better in majority of the datasets.

### 4.3 ONLINE LEARNING

In this section, we investigate whether TAM can handle distribution shifts in online learning, where non-IID setups typically cause deep learning models to struggle due to a loss of plasticity—the ability to adapt to new tasks. In such setups, distribution shifts alter the loss landscape, pushing parameters that performed well on a previous task into sub-optimal, higher loss regions for the new task, leading to plasticity loss (Lewandowski et al., 2024; Elsayed & Mahmood, 2024). Existing solutions to this problem focus on regularization (Kumar et al., 2023), reinitializing inactive parameters (Sokar et al., 2023), or adding normalization layers (Lyle et al., 2024b), often using SGD as the base optimizer.

We hypothesize that TAM's momentum from previous tasks can help push parameters out of sub-optimal regions by mitigating the torqued gradients that arise at the start of the new task, allowing for better exploration of the new task's loss landscape using knowledge from previous gradients. To test this, we compare TAM with SGD and SGDM in an online learning setup. Specifically, similar to (Lyle et al., 2024a;b), we also train multi-layered networks (MLP) on a sequence of tasks, where each task involves image classification on CIFAR10. We induce distribution shifts by flipping the labels between tasks, a common benchmark in online learning research (Elsayed & Mahmood, 2024; Lewandowski et al., 2024). We experiment with different degrees of label flipping, $\delta \in \{40\%, 80\%, 100\%\}$, to simulate soft and hard task boundaries. For each optimizer and each setup, a hyper-parameter grid search is conducted across different effective learning rates, selecting the best-performing setup is selected based on average online accuracy across all tasks, following Dohare et al. (2021). Each task is assigned a compute budget of 40 training epochs. We evaluate on two different sizes of MLP. Further setup details are provided in Appendix A.1.

In Figure 5 (first row), we observe that with smaller MLPs, TAM performs similarly to SGDM across most tasks, with both optimizers consistently outperforming SGD for $\delta = 40\%$. As $\delta$ increases to

---

[1]Performance is considered similar if the difference in scores between AdaTAMW and AdamW is less than 0.2% of the highest score on a given dataset.

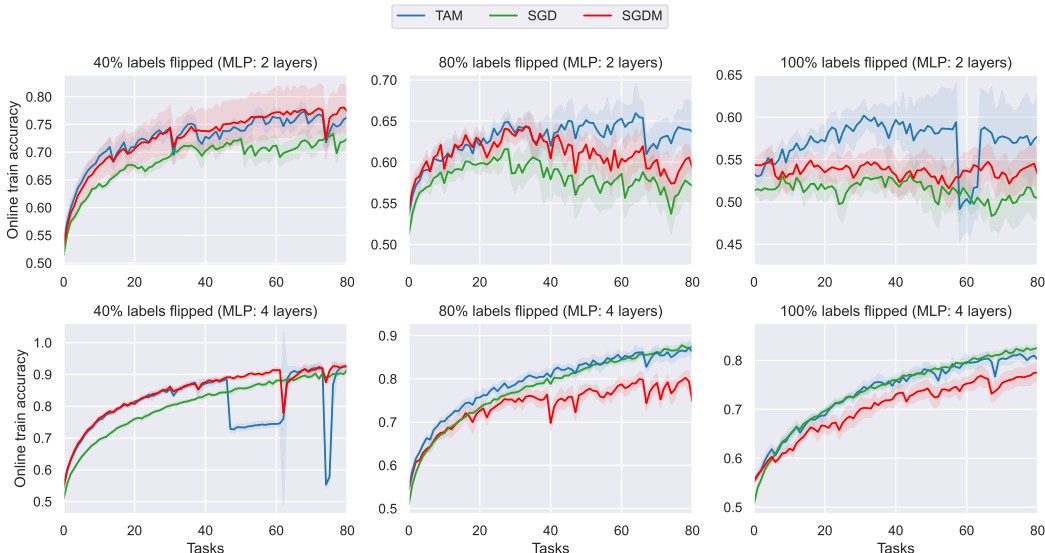

Figure 5: Comparing online accuracy of TAM with SGDM and SGD on label flipping benchmark for training MLP with 2 hidden layers (first row) and 4 hidden layers (second row) after hyper-parameter search across effective learning rates for the following: (i) $40\%$ labels flipping, (ii) $80\%$ labels flipping, and (iii) $100\%$ labels flipping. Although TAM performs similar to SGDM for smoother shifts ($40\%$), it tends to outperform SGDM when distribution shifts are more drastic ($80\%$ and $100\%$).

$80\%$ or $100\%$, TAM outperforms both SGD and SGDM. Notably, for $\delta = 80\%$, TAM maintains higher accuracy and better stability beyond 30 tasks, while SGD and SGDM degrade. At $\delta = 100\%$, TAM continues to show superior accuracy, with a clear gap from the beginning as SGD and SGDM struggle to transfer knowledge for future tasks.

For larger MLPs, TAM performs similarly to SGDM at $\delta = 40\%$, but at higher $\delta$ values, it matches SGD's performance, with both optimizers outperforming SGDM. These results further highlight TAM's robustness, as it not only matches SGDM's adaptability to distribution shifts but also surpasses it in more challenging online learning settings.

## 4.4 WARM-UP WITH TAM

Exploring the loss surface is especially important during the initial phase of training, as it helps the optimizer effectively navigate the loss landscape and avoid getting stuck in local minima. TAM can be beneficial as a warm-up strategy, as it prioritizes important directions, helping to identify the basin of attraction early on.

In this section, we perform an ablation study to evaluate TAM warmup when training a ResNet18 on CIFAR-10. We begin by training the model with TAM and a constant learning rate for a specified number of steps (denoted as $sw$), then switch to SGDM while keeping the effective learning rate and optimizer state same. The learning rate of SGDM is set to half of TAM's learning rate, based on the effective learning rate analysis in section 3. Additionally, we include a baseline where training starts with SGDM, followed by a halving of the learning rate at step $sw$, while maintaining the optimizer state. Further implementation details are provided in Appendix A.1.

In Figure 6 (left), we observe that warmup using TAM leads to higher validation accuracy compared to SGDM for both $sw = 25$ and $sw = 50$. To understand how TAM and SGDM navigate through different regions of the loss landscape, we plot gradient norm (middle) and observe that while an abrupt jump occurs for both methods that could be result of oscillations in sharp minima (Xing et al., 2018). However, this jump is delayed by around $5 - 10$ epochs in TAM suggesting that TAM defers such oscillatory behaviour and explores the landscape for more epochs. We also apply mode connectivity to further analyze the optimization trajectories. Following Frankle et al. (2020), we create two copies of the model at each epoch, train them both until convergence with different order

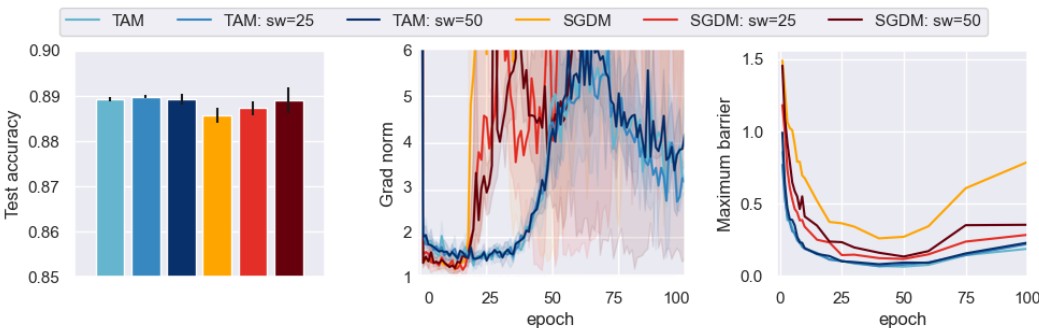

Figure 6: (i) Comparing the performance of TAM and SGDM while training ResNet18 on CIFAR10 with a fixed learning rate across different switching steps ($sw$). Overall, TAM with/without warmup leads to improved validation accuracy compared to SGDM. (ii) Gradient norm observed during training. There is an abrupt jump in gradient norm that occurs first for all SGDM variants (iii) Maximum loss barrier observed during training. Notably, the most significant gain for SGDM warmup occurs at $sw = 50$, which coincides with the lowest observed loss barrier.

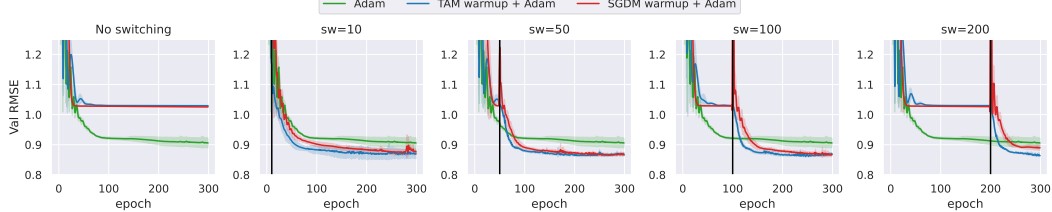

Figure 7: Evaluating warmup with TAM on Link prediction task using GNNs for different switching steps. While warmup with both TAM and SGDM improve the performance for different switching steps, we observed that TAM warmup + Adam has faster convergence speed and result in a lower validation RMSE compared to SGDM warmup + Adam.

of batches such that the models follow different trajectories. We then calculate the loss barrier by interpolating between the weights of converged models. As shown in Figure 6 (right), the maximum barrier starts high, drops significantly until 50 epochs, then increases again. TAM results in relatively lower barriers, indicating greater stability and better connectivity in the loss landscape. Interestingly, the most significant gain for SGDM warmup occurs at $sw = 50$, which coincides with the lowest observed loss barrier, suggesting that mode connectivity can help determine optimal steps even for SGDM warmup.

**Testing warmup on a different loss surface.** We conduct a similar ablation using another architecture to test whether TAM warmup aids in discovering a better region in an other type of loss landscape. Specifically, we train a Graph Neural Network (GNN) to solve a link prediction problem (Harper & Konstan, 2015; Zhang & Chen, 2018), following the open-source implementation [2]. We compare three setups: (i) Adam, the default optimizer used in this setting, (ii) TAM warmup + Adam, and (iii) SGDM warmup + Adam. In the warmup settings, the respective optimizer is used for the first few epochs, then switched to Adam. The models are trained for 300 epochs, and we evaluate the optimizers based on the best validation Root Mean Square Error (RMSE). We test different switching steps ($sw \in \{10, 50, 100, 200\}$), with TAM and SGDM learning rates obtained through a grid search across $\{0.1, 0.01, 0.001\}$ on a held-out dataset. For both TAM warmup and SGDM warmup, $\eta = 0.01$ yields the best results. Adam's learning rate remains fixed at 0.001.

In Figure 7, we plot the validation RMSE for each setup. We also include results with no switching, where the initial optimizer was used for the entire training process. In this setting, Adam outperforms non-adaptive momentum-based methods for the GNN architecture.

[2]Notebook: Link Prediction on MovieLens

The results show that TAM warmup consistently leads to better validation RMSE compared to both naive Adam and SGDM warmup + Adam. Notably, after switching to Adam, the TAM warmup setting exhibits faster convergence than SGDM warmup across all switching steps. The lowest validation RMSE of $0.86$ is achieved with TAM warmup at $sw = 50$ epochs, also suggesting that switching at $sw = 10$ epochs is too early for this particular setup. Additionally, as we increase $sw$, the convergence speed after switching decreases, particularly with SGDM warmup. These findings suggest that TAM, when combined with appropriate warmup steps, can guide the model to a better generalizing region of the loss landscape compared to Adam alone.

## 5    CONCLUSION

We propose Torque-Aware Momentum (TAM), an enhancement of classical momentum that mitigates the detrimental effects of torqued gradients, enabling more stable and consistent exploration of the loss landscape. By incorporating a damping factor that adjusts momentum based on gradient alignments, TAM helps models escape sharp minima and improve generalization across diverse tasks.

Our evaluation of TAM spans multiple experimental setups, including image classification, large language model fine-tuning, and online learning with distribution shifts. Across these tasks, TAM consistently performs on par with, and often surpasses, traditional SGD and SGDM. In particular, TAM shows significant advantages in tasks involving distribution shifts, where it stabilizes learning and adapts more effectively than SGDM, especially when tasks share little overlap. Additionnaly, TAM proves valuable as a warm-up strategy, leading to faster convergence and lower loss barriers compared to SGDM.

While our results demonstrate TAM's effectiveness in tasks with distribution shifts and gradient misalignment, further work is needed to test its capabilites in more challenging non-stationary environments, such as continual learning. Our preliminary continual learning experiments in Appendix A.2.3 highlight TAM's potential to address catastrophic forgetting by retaining gradient direction from previous tasks. However, a thorough investigation is required to fully understand and optimize TAM's performance in this domain. Another exciting avenue is to explore TAM's potential in other training paradigms, such as self-supervised learning and reinforcement learning, where effective exploration and stability is critical for model success.

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

## A  APPENDIX

In this section, we provide the details and results not present in the main content. We describe the implementation details including hyper-parameters values used in our experiments in section A.1. All experiments were executed on an NVIDIA A100 Tensor Core GPUs machine with 40 GB memory.

### A.1  IMPLEMENTATION DETAILS

#### A.1.1  DATASETS AND MODELS

| Dataset | Train set | Validation set |
|---------|-----------|----------------|
| CIFAR10 | 40K | 10K |
| CIFAR100 | 40K | 10K |
| ImageNet | 1281K | 50K |
| MovieLens | 80K | 10K |

Table 2: Dataset details

In Table 2 and Table 3, we provide a summary of all datasets and models used in image classification (subsection 4.1), LLM experiments (subsection 4.2) except details on MTEB which is later described in subsubsection A.2.7, online learning (subsection 4.3) and GNN experiments (Figure 7).

| Model | Number of parameters | Other details |
|-------|---------------------|---------------|
| MobileNet | 13M | |
| ResNet18 | 11M | |
| ResNet34 | 22M | |
| ResNet50 | 25.5M | |
| ViT | 87M | |
| BERT-base | 110M | 12-layers, 768-hidden |
| BERT-large | 340M | 24-layers, 1024-hidden |
| DeBERTa-base | 86M | 12-layers, 768-hidden |
| DeBERTa-large | 304M | 24-layers, 1024-hidden |
| RoBERTa-base | 125M | 12-layers, 768-hidden |
| RoBERTa-large | 355M | 24-layers, 1024-hidden |
| MLP-2 | 412K | 2-layers, 128-hidden |
| MLP-4 | 460K | 4-layers, 128-hidden |
| GNN | 80K | 2-layers |

Table 3: Model details

#### A.1.2  HYPER-PARAMETERS

Unless specified in the experiment description, the default set of hyperparameters in all our experiments is for momentum-based methods are $\{\eta, \beta_1\} = \{0.1, 0.9\}$ and similarly for adaptive optimizers are $\{\eta, \beta_1, \beta_2\} = \{0.001, 0.9, 0.999\}$.

For image classification and online learning experiments, we provide the details on hyper-parameter grid-search in Table 4 and the best settings for all experiments and Table 5.

### A.2  ADDITIONAL RESULTS

#### A.2.1  CONNECTION WITH SGDM CONVERGENCE

In Figure 8, we plot $\hat{s}_t$ from Eq. 3 obtained during training ResNet18 on CIFAR10/100. We observe that after $\hat{s}_t$ has a positive value at the start, then it fluctuates and drops to a negative value and eventually increases and saturates near $s^* = 0$ in both cases.

| Optimizer | Learning rate set |
|-----------|-------------------|
| SGD | $\{0.1, 0.01, 0.001, 0.0001\}$ |
| SGDM | $\{0.1, 0.01, 0.001, 0.0001\}$ |
| TAM | $\{0.2, 0.02, 0.002, 0.0002\}$ |
| Adam | $\{0.1, 0.01, 0.001, 0.0001\}$ |
| AdaTAM | $\{0.1, 0.01, 0.001, 0.0001\}$ |
| AngularGrad | $\{0.1, 0.01, 0.001, 0.0001\}$ |
| Online SGD | $\{0.005, 0.01, 0.02, 0.03\}$ |
| Online SGDM | $\{0.005, 0.01, 0.02, 0.03\}$ |
| Online TAM | $\{0.01, 0.02, 0.04, 0.06\}$ |

Table 4: Details on grid search on image classification and online learning experiment.

| Optimizers | CIFAR10 | CIFAR10 | CIFAR100 | CIFAR100 | ImageNet | Shuffled CIFAR10 | Shuffled CIFAR10 |
|------------|---------|---------|----------|----------|----------|------------------|------------------|
| | **ResNet18** | **ResNet34** | **ResNet18** | **ResNet34** | **ResNet50** | **MLP-2** | **MLP-4** |
| SGD | 0.1 | 0.1 | 0.1 | 0.1 | 0.1 | $\{0.03, 0.03, 0.03\}$ | $\{0.03, 0.03, 0.03\}$ |
| SGDM | 0.1 | 0.01 | 0.01 | 0.1 | 0.1 | $\{0.02, 0.01, 0.01\}$ | $\{0.005, 0.005, 0.005\}$ |
| TAM | 0.2 | 0.2 | 0.2 | 0.2 | 0.2 | $\{0.04, 0.02, 0.04\}$ | $\{0.01, 0.01, 0.02\}$ |
| Adam | 0.001 | 0.001 | 0.001 | 0.001 | 0.0001 | – | – |
| AngularGrad | 0.001 | 0.001 | 0.001 | 0.001 | 0.0001 | – | – |
| AdaTAM | 0.0001 | 0.0001 | 0.0001 | 0.0001 | 0.0001 | – | – |

Table 5: Best learning rate for different optimizers on image classification and online learning benchmarks.

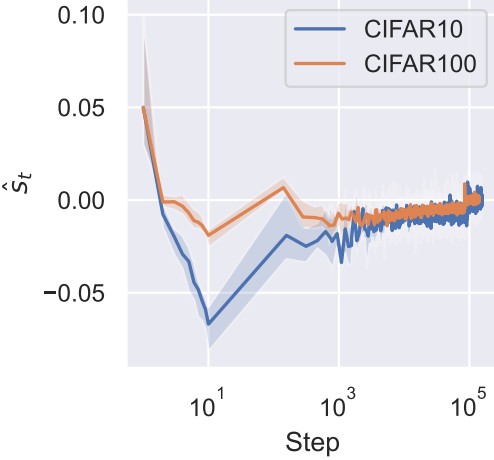

Figure 8: Evolution of $\hat{s}_t$ during training ResNet18 on CIFAR10 and CIFAR100 datasets. We observe that after starting from positive value, $\hat{s}_t$ drops to negative, fluctuates and eventually saturates near $s^* = 0$ in both cases.

### A.2.2 IMAGE CLASSIFICATION

We evaluate TAM on the following models: (i) MobileNet (Howard, 2017): Similar to ResNet experiments described in subsection 4.1, we train MobileNet for 200 epochs and perform the learning rate grid search (Table 4) to obtain the best setup. (ii) Vision Transformer (ViT) (Dosovitskiy et al., 2021): We fine-tune a ViT model on CIFAR10/100 that was pre-trained on ImageNet dataset.[3] For SGDM and SGD on ViT, we select the learning rate from the grid $\{0.01, 0.03, 0.1, 0.3\}$, whereas for TAM, we choose it from $\{0.02, 0.06, 0.2, 0.6\}$. The resulting test accuracy along with the best learning rates are reported in Table 6. We observe that TAM and AdaTAM either match the performance of SGDM or outperform the baselines.

---

[3]Notebook: Vision Transformer

| Model | Optimizers | CIFAR10 | | CIFAR100 | |
|-------|-----------|---------|---------|----------|---------|
| | | Accuracy | Best LR | Accuracy | Best LR |
| **MobileNet** | SGD | $93.7_{\pm 0.2}$ | 0.1 | $72.8_{\pm 0.1}$ | 0.1 |
| | SGDM | $\mathbf{93.9}_{\pm 0.1}$ | 0.01 | $72.8_{\pm 0.3}$ | 0.01 |
| | TAM (ours) | $\mathbf{93.9}_{\pm 0.2}$ | 0.02 | $72.8_{\pm 0.1}$ | 0.02 |
| **MobileNet** | Adam | $92.7_{\pm 0.1}$ | 0.01 | $69.8_{\pm 0.1}$ | 0.001 |
| | AngularGrad | $91.7_{\pm 0.1}$ | 0.001 | $66.4_{\pm 0.1}$ | 0.001 |
| | AdaTAM (ours) | $\mathbf{93.1}_{\pm 0.1}$ | 0.0001 | $\mathbf{70.7}_{\pm 0.3}$ | 0.0001 |
| **ViT fine-tuning** | SGD | $97.1_{\pm 0.1}$ | 0.1 | $74.4_{\pm 0.6}$ | 0.03 |
| | SGDM | $\mathbf{97.7}_{\pm 0.1}$ | 0.1 | $85.3_{\pm 0.2}$ | 0.1 |
| | TAM (ours) | $\mathbf{97.7}_{\pm 0.1}$ | 0.2 | $\mathbf{86.2}_{\pm 0.2}$ | 0.2 |

Table 6: Comparison of TAM and AdaTAM with baseline optimizers for MobileNet and ViT trained on CIFAR10/100 with learning rate grid search.

### A.2.3 CONTINUAL LEARNING

In subsection 4.3, we evaluate TAM on an online learning setup where we showed that TAM helps in maintaining the plasticity of MLP across a large number of tasks. In this section, we evaluate TAM in a more challenging setting - continual learning - where the goal is to maintain both the stability and plasticity of the model. In particular, we train a ResNet50 model on CLEAR benchmark (Lin et al., 2021) which consists of 10 sequential image recognition tasks (or experiences) with the goal of maximizing average accuracy on all tasks without forgetting. We follow the implementation provided by Zhang et al. (2023) to compare SGDM and TAM optimizers. We evaluate these two optimizers on top of two continual learning setups: Naive and Learning without forgetting (LwF) (Li & Hoiem, 2017) which is a well-known continual learning method. We conduct a grid search across learning rate (from set $\{0.005, 0.1, 0.2\}$) and select the best setup based on performance on a held-out dataset. The learning rate of $0.005$ performed best for both SGDM and TAM.

In Table 7, we report the accuracies obtained on the evaluation set of each experience when the model was sequentially trained on all tasks. Overall, we observe that under both setups, TAM outperforms SGDM in all experiences. Interestingly, in some cases, TAM with Naive setup also performs better than SGDM with LwF. These results suggest that TAM can maintain both stability and plasticity better than SGDM.

| Methods | Optimizers | $Exp_1$ | $Exp_2$ | $Exp_3$ | $Exp_4$ | $Exp_5$ | $Exp_6$ | $Exp_7$ | $Exp_8$ | $Exp_9$ | $Exp_{10}$ |
|---------|-----------|-------|-------|-------|-------|-------|-------|-------|-------|-------|--------|
| **Naive** | SGDM | 89.1 | 90.1 | 89.8 | 89.4 | 92.0 | 90.7 | 90.4 | 91.2 | 89.9 | 93.7 |
| | TAM (ours) | 90.9 | 90.5 | 92.5 | 92.2 | 93.6 | 92.7 | 92.4 | 92.9 | 93.4 | 95.9 |
| **LwF** | SGDM | 93.2 | 93.3 | 93.7 | 93.8 | 94.0 | 92.3 | 93.4 | 94.5 | 93.3 | 96.1 |
| | TAM (ours) | 95.3 | 94.6 | 94.6 | 94.5 | 97.1 | 94.4 | 94.8 | 95.3 | 94.5 | 96.3 |

Table 7: Comparing final accuracy(%) obtained using TAM and SGDM on the evaluation set of each CLEAR dataset experience under Naive and LwF setups in continual learning. We observe that in both setups, TAM outperforms SGDM on all experiences.

### A.2.4 VARYING $\gamma$

In this section, we conduct a brief ablation study on ResNet18 to compare the effects of varying $\gamma$ (in Eq. 3) while keeping all other hyperparameters fixed for CIFAR10 and CIFAR100. The results are reported in Table 8. We observe that varying gamma has minimal impact on the overall behavior of the optimization trajectories and therefore, even with changes in gamma, TAM consistently outperforms other baselines.

| Dataset | $\gamma$ | TAM |
|---|---|---|
| **CIFAR10** | 0.99 | $93.9_{\pm 0.1}$ |
| | 0.9 | $94.2_{\pm 0.2}$ (reported in Table 1) |
| | 0.8 | $94.1_{\pm 0.2}$ |
| | 0.5 | $93.9_{\pm 0.1}$ |
| | 0.0 | $94.1_{\pm 0.1}$ |
| **CIFAR100** | 0.99 | $74.0_{\pm 0.1}$ |
| | 0.9 | $73.8_{\pm 0.1}$ (reported in Table 1) |
| | 0.8 | $74.1_{\pm 0.3}$ |
| | 0.5 | $73.7_{\pm 0.4}$ |
| | 0.0 | $74.1_{\pm 0.3}$ |

Table 8: Performance comparison for different $\gamma$ values for training ResNet18 on CIFAR10 and CIFAR100. Varying gamma has minimal impact on the overall behavior of the optimization trajectories.

### A.2.5   AdaTAM with exponential moving average

In this experiment, we consider an alternate update rule for momentum as compared to Eq. 7 as follows:

$$m_t = (1 - (\epsilon + d_t))m_{t-1} + (\epsilon + d_t)g_t \ . \tag{8}$$

Specifically, the above update rule uses an exponential moving average to update momentum. We call this variant AdaTAM2 and compare its performance with the default AdaTAM in Table 9. We observe that incorporating an exponential moving average into AdaTAM had minimal impact on performance and, on CIFAR100, it slightly degraded it.

| | CIFAR10 | | CIFAR100 | |
|---|---|---|---|---|
| **Metric** | **ResNet18** | **ResNet34** | **ResNet18** | **ResNet34** |
| AdaTAM (Reported in Table 1) | $93.3_{\pm 0.3}$ | $93.3_{\pm 0.1}$ | $\mathbf{72.7}_{\pm 0.3}$ | $\mathbf{72.9}_{\pm 0.1}$ |
| AdaTAM2 | $93.3_{\pm 0.1}$ | $\mathbf{93.6}_{\pm 0.2}$ | $71.9_{\pm 0.2}$ | $72.6_{\pm 0.1}$ |

Table 9: Performance comparison of AdaTAM and its variant AdaTAM2 which uses exponential moving average to update momentum on CIFAR10 and CIFAR100 with ResNet18 and ResNet34.

### A.2.6   Warm-up with TAM

We conducted a gradient norm analysis similar to that shown in Figure 6 and present the results in Figure 9:

1. On SVHN (Netzer et al., 2011), we observe a pattern similar to CIFAR10 in Figure 6 (second). Both SGDM and TAM exhibit abrupt jumps in gradient norm, possibly due to oscillations in sharp minima. However, TAM defers this oscillatory behavior and explores the loss landscape for more epochs, showcasing its ability to maintain stability for a longer period during training.

2. On CIFAR100, we observe that TAM avoids abrupt jumps in gradient norm during the first 100 epochs. Moreover, SGDM with $sw = 25$ also demonstrates controlled gradient norms, indicating improved training stability.

3. On comparing AdaTAM with Adam on CIFAR100, the results indicate that AdaTAM consistently maintains a lower gradient norm as training progresses whereas the gradient norm in Adam decreases gradually over time. This suggests that the damping effect in AdaTAM effectively controls large gradients.

### A.2.7   LLM Fine-tuning

Figure 10 shows the percentage of times AdaTAMW performed similarly or better than AdamW. Unlike Figure 4, the performance is considered similar if the difference in scores between AdaTAMW

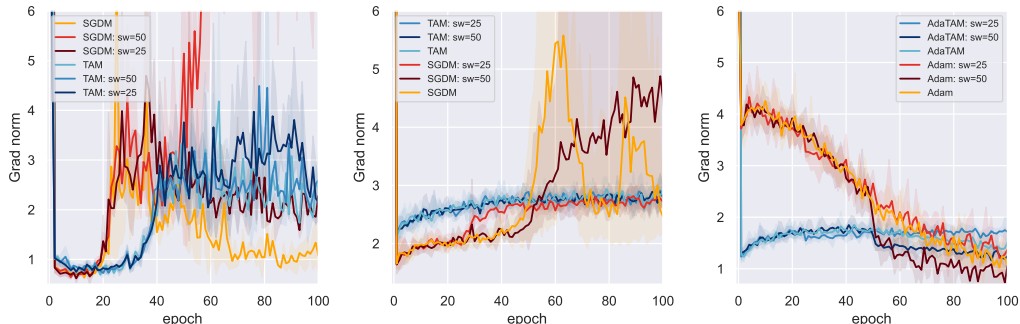

Figure 9: Comparing gradient norm observed during training ResNet18 similar to Figure 6 for (i) TAM vs SGDM on SVHN, (ii) TAM vs SGDM on CIFAR100 and (iii) AdaTAM vs Adam on CIFAR100. There is an abrupt jump in gradient norm that occurs first for SGDM variants whereas for training with Adam, gradient norm gradually decreases from a higher value. In case of TAM on SVHN, the abrupt jump is delayed by few epochs as compared to SGDM. On CIFAR100, both TAM and AdaTAM maintain a low gradient norm for the first 100 epochs.

and AdamW is less than $1\%$ of the highest score on a given dataset. Except for the BERT-base fine-tuned on 10 epochs, AdaTAMW generally matches or exceeds AdamW's performance in most settings.

|  | 3 | 5 | 10 |
|---|---|---|---|
| BERT-base | 71% (100%) | 50% (96%) | 18% (45%) |
| BERT-large | 68% (73%) | 75% (77%) | 48% (52%) |
| DeBERTa-base | 54% (88%) | 43% (79%) | 64% (93%) |
| DeBERTa-large | 93% (95%) | 73% (79%) | 50% (61%) |
| RoBERTa-base | 80% (96%) | 84% (95%) | 88% (98%) |
| RoBERTa-large | 39% (55%) | 43% (52%) | 39% (55%) |

epochs

Figure 10: Percentage of times when AdaTAMW performs better (or similar/better) than AdamW on various LLMs across 56 MTEB datasets. Green indicates that AdaTAMW achieves similar or better performance, while red indicates worse performance. Except for BERT models with 10 epochs and RoBERTa-large, AdaTAMW performs similar/better in majority of the datasets.

### A.2.8 RUNTIME

In terms of runtime, since AdaTAMW only introduces computation overhead of cosine similarity, it is only 1.12x slower than runtime of AdamW. For example, we provide time spent on finetuning BERT-large model in Table 10.

### A.2.9 DETAILED MTEB RESULTS

We report the exact scores obtained on all 56 MTEB datasets for all types of BERT model in Table 11, Table 12, Table 13, Table 14, Table 15 and Table 16.

| Epochs | AdaTAMW | AdamW |
|--------|---------|-------|
| 3      | 19.75   | 17.58 |
| 5      | 32.00   | 28.67 |
| 10     | 63.83   | 57.00 |

Table 10: Running time (in minutes) comparison for AdaTAMW and AdamW on finetuning BERT-large on Wikitext dataset for different number of epochs.

| | AdamW/3 | AdamW/5 | AdamW/10 | AdaTAMW/3 | AdaTAMW/5 | AdaTAMW/10 |
|---|---|---|---|---|---|---|
| AmazonCounterfactualClassification | 68.77 | 68.86 | 68.68 | 68.71 | 68.82 | 68.69 |
| AmazonPolarityClassification | 70.24 | 70.17 | 70.70 | 70.26 | 70.18 | 70.51 |
| AmazonReviewsClassification | 26.14 | 26.14 | 26.15 | 26.15 | 26.13 | 26.15 |
| Banking77Classification | 56.94 | 57.76 | 57.52 | 57.80 | 57.25 | 47.81 |
| EmotionClassification | 34.79 | 34.70 | 34.89 | 34.82 | 34.74 | 35.00 |
| ImdbClassification | 63.84 | 63.81 | 64.31 | 63.85 | 63.83 | 63.84 |
| MTOPDomainClassification | 53.66 | 53.69 | 53.58 | 53.70 | 53.65 | 53.66 |
| MTOPIntentClassification | 40.20 | 40.20 | 40.17 | 40.19 | 40.14 | 35.20 |
| MassiveIntentClassification | 29.90 | 29.26 | 29.11 | 29.93 | 29.86 | 29.36 |
| MassiveScenarioClassification | 31.39 | 31.29 | 31.07 | 31.40 | 31.28 | 31.39 |
| ToxicConversationsClassification | 67.72 | 67.61 | 67.13 | 67.65 | 67.58 | 67.30 |
| TweetSentimentExtractionClassification | 50.39 | 50.40 | 50.48 | 50.39 | 50.45 | 50.61 |
| ArxivClusteringP2P | 34.14 | 34.37 | 34.47 | 34.20 | 34.36 | 34.40 |
| ArxivClusteringS2S | 25.89 | 25.94 | 26.23 | 25.96 | 25.99 | 26.03 |
| BiorxivClusteringP2P | 28.07 | 28.43 | 28.69 | 28.01 | 28.41 | 28.37 |
| BiorxivClusteringS2S | 22.03 | 22.10 | 22.47 | 22.07 | 22.10 | 22.13 |
| MedrxivClusteringP2P | 24.91 | 25.02 | 25.33 | 24.90 | 24.96 | 25.00 |
| MedrxivClusteringS2S | 22.04 | 21.97 | 22.12 | 21.89 | 22.04 | 22.11 |
| RedditClustering | 22.20 | 22.96 | 24.21 | 22.28 | 22.92 | 22.94 |
| RedditClusteringP2P | 41.24 | 41.55 | 42.13 | 41.43 | 41.84 | 41.58 |
| StackExchangeClustering | 39.65 | 40.03 | 41.14 | 39.63 | 39.91 | 40.21 |
| StackExchangeClusteringP2P | 25.74 | 25.73 | 26.00 | 25.75 | 25.80 | 25.84 |
| TwentyNewsgroupsClustering | 18.50 | 18.79 | 20.43 | 18.75 | 18.64 | 18.97 |
| SprintDuplicateQuestions | 41.99 | 42.02 | 43.01 | 42.06 | 41.71 | 41.90 |
| TwitterSemEval2015 | 57.80 | 57.84 | 57.94 | 57.81 | 57.77 | 57.83 |
| TwitterURLCorpus | 76.06 | 76.25 | 76.62 | 76.07 | 76.22 | 76.09 |
| AskUbuntuDupQuestions | 47.74 | 47.86 | 48.01 | 47.71 | 47.81 | 47.41 |
| MindSmallReranking | 26.98 | 27.09 | 27.32 | 27.00 | 27.08 | 26.90 |
| SciDocsRR | 62.05 | 62.18 | 62.67 | 62.09 | 62.20 | 62.32 |
| StackOverflowDupQuestions | 36.51 | 36.32 | 36.33 | 36.51 | 36.43 | 36.39 |
| ArguAna | 28.24 | 28.47 | 28.72 | 28.29 | 28.55 | 28.20 |
| CQADupstackTexRetrieval | 4.30 | 4.46 | 4.86 | 4.32 | 4.46 | 4.32 |
| ClimateFEVER | 7.56 | 7.90 | 8.34 | 7.59 | 8.01 | 7.78 |
| DBPedia | 7.10 | 7.36 | 7.97 | 7.07 | 7.34 | 7.52 |
| FEVER | 6.29 | 6.45 | 7.56 | 6.25 | 6.62 | 6.50 |
| FiQA2018 | 3.83 | 4.05 | 4.44 | 3.91 | 4.02 | 4.07 |
| HotpotQA | 9.77 | 9.66 | 9.70 | 9.76 | 9.61 | 9.62 |
| MSMARCO | 3.76 | 3.91 | 4.03 | 3.79 | 3.91 | 3.82 |
| NFCorpus | 7.52 | 7.41 | 7.51 | 7.48 | 7.32 | 7.40 |
| NQ | 5.48 | 5.58 | 5.99 | 5.54 | 5.61 | 5.45 |
| QuoraRetrieval | 61.45 | 61.55 | 62.15 | 61.43 | 61.57 | 61.58 |
| SCIDOCS | 4.44 | 4.45 | 4.57 | 4.49 | 4.42 | 4.46 |
| SciFact | 17.19 | 17.64 | 18.66 | 17.45 | 17.51 | 17.61 |
| TRECCOVID | 16.88 | 17.53 | 19.18 | 16.84 | 17.87 | 16.97 |
| Touche2020 | 3.49 | 3.91 | 4.42 | 3.47 | 3.73 | 3.65 |
| BIOSSES | 55.56 | 55.38 | 54.58 | 55.69 | 54.89 | 55.17 |
| SICK-R | 60.52 | 60.66 | 60.99 | 60.54 | 60.74 | 60.75 |
| STS12 | 33.51 | 34.45 | 36.26 | 33.56 | 34.54 | 33.92 |
| STS13 | 60.04 | 60.20 | 61.28 | 60.09 | 60.28 | 60.04 |
| STS14 | 48.89 | 49.38 | 50.76 | 48.93 | 49.41 | 49.03 |
| STS15 | 63.48 | 63.79 | 64.93 | 63.51 | 63.88 | 63.38 |
| STS16 | 63.22 | 63.40 | 64.10 | 63.16 | 63.49 | 63.09 |
| STS17 | 21.05 | 20.99 | 20.76 | 21.08 | 20.99 | 21.03 |
| STS22 | 19.77 | 20.77 | 21.45 | 19.93 | 20.80 | 21.35 |
| STSBenchmark | 51.21 | 52.18 | 53.80 | 51.26 | 52.36 | 51.54 |
| SummEval | 29.61 | 29.85 | 30.03 | 29.62 | 29.89 | 29.82 |

Table 11: Performance on all 56 MTEB datasets obtained on BERT-base.

| | AdamW/3 | AdamW/5 | AdamW/10 | AdaTAMW/3 | AdaTAMW/5 | AdaTAMW/10 |
|---|---|---|---|---|---|---|
| AmazonCounterfactualClassification | 65.25 | 64.78 | 67.17 | 67.38 | 65.52 | 67.01 |
| AmazonPolarityClassification | 67.98 | 67.45 | 68.03 | 69.90 | 67.46 | 71.12 |
| AmazonReviewsClassification | 24.06 | 24.02 | 24.75 | 25.29 | 24.33 | 25.15 |
| Banking77Classification | 47.59 | 43.45 | 47.71 | 46.04 | 43.66 | 45.85 |
| EmotionClassification | 27.00 | 25.10 | 28.19 | 27.77 | 26.65 | 27.00 |
| ImdbClassification | 66.21 | 66.07 | 64.68 | 65.58 | 64.93 | 66.79 |
| MTOPDomainClassification | 41.55 | 39.64 | 45.69 | 45.92 | 40.75 | 43.04 |
| MTOPIntentClassification | 26.89 | 28.04 | 31.25 | 32.00 | 25.73 | 28.33 |
| MassiveIntentClassification | 19.29 | 20.31 | 24.95 | 25.47 | 21.14 | 22.53 |
| MassiveScenarioClassification | 20.90 | 22.19 | 26.71 | 26.73 | 22.64 | 24.16 |
| ToxicConversationsClassification | 64.19 | 63.01 | 65.17 | 64.58 | 63.50 | 64.66 |
| TweetSentimentExtractionClassification | 45.75 | 45.67 | 48.40 | 49.45 | 46.97 | 47.54 |
| ArxivClusteringP2P | 34.84 | 35.18 | 32.77 | 33.37 | 33.62 | 35.73 |
| ArxivClusteringS2S | 14.89 | 13.19 | 22.26 | 22.58 | 17.72 | 19.68 |
| BiorxivClusteringP2P | 29.59 | 29.70 | 27.17 | 28.59 | 28.46 | 29.92 |
| BiorxivClusteringS2S | 14.32 | 9.35 | 18.03 | 16.59 | 13.95 | 15.31 |
| MedrxivClusteringP2P | 25.29 | 25.20 | 23.59 | 24.25 | 24.43 | 25.57 |
| MedrxivClusteringS2S | 17.27 | 14.50 | 19.28 | 19.00 | 16.84 | 17.56 |
| RedditClustering | 8.46 | 8.54 | 12.30 | 12.68 | 9.58 | 11.56 |
| RedditClusteringP2P | 31.51 | 32.42 | 28.52 | 31.34 | 28.96 | 34.87 |
| StackExchangeClustering | 22.80 | 19.71 | 27.92 | 27.39 | 23.63 | 25.69 |
| StackExchangeClusteringP2P | 24.10 | 23.96 | 22.96 | 23.24 | 23.53 | 24.21 |
| TwentyNewsgroupsClustering | 9.65 | 9.34 | 12.24 | 11.97 | 10.02 | 10.99 |
| SprintDuplicateQuestions | 29.19 | 16.33 | 38.96 | 37.44 | 32.34 | 35.59 |
| TwitterSemEval2015 | 41.90 | 38.98 | 47.80 | 47.41 | 40.84 | 43.90 |
| TwitterURLCorpus | 53.78 | 50.15 | 65.24 | 67.26 | 56.38 | 58.87 |
| AskUbuntuDupQuestions | 45.72 | 43.96 | 46.88 | 47.17 | 46.11 | 46.31 |
| MindSmallReranking | 25.08 | 25.09 | 26.02 | 26.23 | 24.91 | 25.28 |
| SciDocsRR | 45.57 | 41.86 | 54.45 | 53.62 | 46.51 | 49.32 |
| StackOverflowDupQuestions | 31.80 | 28.20 | 35.52 | 35.23 | 33.10 | 33.46 |
| ArguAna | 22.07 | 23.11 | 17.84 | 18.32 | 18.88 | 22.60 |
| CQADupstackTexRetrieval | 2.84 | 1.82 | 3.07 | 3.37 | 2.62 | 3.26 |
| ClimateFEVER | 6.91 | 7.63 | 4.97 | 6.08 | 6.32 | 7.22 |
| DBPedia | 2.87 | 3.46 | 2.57 | 3.53 | 2.84 | 4.96 |
| FEVER | 5.40 | 3.16 | 3.82 | 5.59 | 4.75 | 6.84 |
| FiQA2018 | 3.56 | 2.37 | 2.80 | 3.09 | 2.81 | 3.73 |
| HotpotQA | 10.68 | 9.13 | 8.57 | 8.18 | 8.57 | 9.89 |
| MSMARCO | 2.82 | 0.63 | 1.98 | 2.25 | 2.13 | 2.08 |
| NFCorpus | 2.72 | 3.42 | 4.01 | 4.46 | 3.27 | 4.08 |
| NQ | 4.81 | 3.48 | 3.43 | 3.51 | 3.69 | 5.47 |
| QuoraRetrieval | 50.94 | 42.47 | 53.34 | 52.94 | 50.21 | 50.09 |
| SCIDOCS | 2.87 | 1.77 | 3.03 | 3.38 | 2.66 | 3.39 |
| SciFact | 16.10 | 16.04 | 15.29 | 15.63 | 16.51 | 21.80 |
| TRECCOVID | 16.76 | 13.33 | 16.05 | 16.81 | 16.38 | 16.84 |
| Touche2020 | 2.95 | 1.95 | 2.55 | 2.63 | 2.27 | 3.06 |
| BIOSSES | 39.02 | 37.44 | 41.46 | 34.51 | 40.96 | 44.87 |
| SICK-R | 36.85 | 34.65 | 40.25 | 39.28 | 35.42 | 39.27 |
| STS12 | 20.27 | 17.60 | 23.65 | 22.08 | 17.70 | 21.67 |
| STS13 | 18.79 | 19.54 | 32.18 | 31.75 | 18.78 | 25.39 |
| STS14 | 22.78 | 19.02 | 30.08 | 29.98 | 21.24 | 26.13 |
| STS15 | 32.86 | 22.31 | 39.90 | 36.49 | 27.56 | 34.74 |
| STS16 | 39.88 | 32.49 | 40.74 | 41.38 | 36.72 | 40.88 |
| STS17 | 13.46 | 14.42 | 14.97 | 14.23 | 14.74 | 15.72 |
| STS22 | 13.62 | 11.94 | 11.18 | 12.66 | 12.56 | 14.99 |
| STSBenchmark | 29.31 | 22.17 | 32.03 | 30.96 | 25.29 | 31.10 |
| SummEval | 30.37 | 30.22 | 30.37 | 30.35 | 30.63 | 31.02 |

Table 12: Performance on all 56 MTEB datasets obtained on BERT-large.

| | AdamW/3 | AdamW/5 | AdamW/10 | AdaTAMW/3 | AdaTAMW/5 | AdaTAMW/10 |
|---|---|---|---|---|---|---|
| AmazonCounterfactualClassification | 67.79 | 68.40 | 69.11 | 67.96 | 68.43 | 68.85 |
| AmazonPolarityClassification | 63.98 | 66.05 | 67.98 | 63.98 | 65.98 | 67.50 |
| AmazonReviewsClassification | 28.93 | 30.09 | 30.72 | 28.93 | 30.09 | 30.64 |
| Banking77Classification | 39.95 | 41.65 | 43.62 | 38.67 | 40.06 | 43.59 |
| EmotionClassification | 25.28 | 26.58 | 27.56 | 25.14 | 26.72 | 27.81 |
| ImdbClassification | 60.39 | 62.45 | 64.58 | 60.48 | 62.62 | 64.64 |
| MTOPDomainClassification | 48.02 | 49.53 | 51.61 | 47.79 | 49.47 | 51.94 |
| MTOPIntentClassification | 32.65 | 34.00 | 35.66 | 32.38 | 33.90 | 35.83 |
| MassiveIntentClassification | 18.75 | 19.41 | 21.54 | 18.69 | 19.83 | 21.76 |
| MassiveScenarioClassification | 22.81 | 23.79 | 25.57 | 22.98 | 23.88 | 25.83 |
| ToxicConversationsClassification | 60.92 | 61.13 | 62.51 | 60.92 | 61.05 | 62.00 |
| TweetSentimentExtractionClassification | 46.65 | 48.28 | 49.81 | 46.68 | 48.45 | 49.82 |
| ArxivClusteringP2P | 22.04 | 23.46 | 23.94 | 21.92 | 23.45 | 24.19 |
| ArxivClusteringS2S | 14.47 | 15.70 | 15.88 | 14.50 | 15.61 | 15.99 |
| BiorxivClusteringP2P | 16.57 | 18.81 | 20.39 | 16.55 | 18.80 | 20.55 |
| BiorxivClusteringS2S | 10.16 | 11.51 | 12.16 | 10.12 | 11.41 | 12.33 |
| MedrxivClusteringP2P | 19.48 | 20.67 | 21.42 | 19.42 | 20.66 | 21.59 |
| MedrxivClusteringS2S | 17.46 | 18.09 | 18.44 | 17.50 | 18.22 | 18.42 |
| RedditClustering | 14.10 | 15.42 | 15.97 | 14.13 | 15.34 | 15.98 |
| RedditClusteringP2P | 27.59 | 29.70 | 31.50 | 27.78 | 29.77 | 31.43 |
| StackExchangeClustering | 22.08 | 24.29 | 25.81 | 21.93 | 24.23 | 25.85 |
| StackExchangeClusteringP2P | 26.83 | 27.10 | 27.10 | 26.93 | 27.12 | 27.26 |
| TwentyNewsgroupsClustering | 12.97 | 13.91 | 14.06 | 13.09 | 13.72 | 13.98 |
| SprintDuplicateQuestions | 15.46 | 18.71 | 21.50 | 15.46 | 18.93 | 21.60 |
| TwitterSemEval2015 | 50.32 | 50.48 | 51.25 | 50.30 | 50.42 | 51.05 |
| TwitterURLCorpus | 65.46 | 66.17 | 66.68 | 65.34 | 66.20 | 66.57 |
| AskUbuntuDupQuestions | 43.89 | 44.37 | 45.02 | 44.22 | 44.33 | 44.93 |
| MindSmallReranking | 26.97 | 27.41 | 27.60 | 27.06 | 27.46 | 27.62 |
| SciDocsRR | 43.55 | 45.39 | 46.76 | 43.58 | 45.30 | 46.70 |
| StackOverflowDupQuestions | 30.19 | 30.92 | 31.75 | 30.13 | 30.89 | 31.74 |
| ArguAna | 8.95 | 11.46 | 12.88 | 9.03 | 11.32 | 13.24 |
| CQADupstackTexRetrieval | 0.35 | 0.47 | 0.78 | 0.34 | 0.49 | 0.82 |
| ClimateFEVER | 0.55 | 0.89 | 1.30 | 0.47 | 0.82 | 1.09 |
| DBPedia | 0.10 | 0.14 | 0.25 | 0.10 | 0.14 | 0.30 |
| FEVER | 0.12 | 0.16 | 0.58 | 0.10 | 0.16 | 0.47 |
| FiQA2018 | 0.31 | 0.33 | 0.44 | 0.31 | 0.29 | 0.43 |
| HotpotQA | 0.22 | 0.40 | 0.77 | 0.23 | 0.39 | 0.92 |
| MSMARCO | 0.05 | 0.08 | 0.18 | 0.04 | 0.09 | 0.19 |
| NFCorpus | 1.45 | 1.59 | 1.68 | 1.43 | 1.56 | 1.70 |
| NQ | 0.04 | 0.06 | 0.15 | 0.04 | 0.08 | 0.16 |
| QuoraRetrieval | 32.25 | 35.91 | 39.37 | 32.47 | 36.01 | 39.61 |
| SCIDOCS | 0.21 | 0.28 | 0.39 | 0.21 | 0.27 | 0.39 |
| SciFact | 2.33 | 3.40 | 5.15 | 2.30 | 3.28 | 5.58 |
| TRECCOVID | 5.11 | 5.46 | 5.25 | 5.75 | 5.33 | 6.23 |
| Touche2020 | 0.13 | 0.53 | 0.64 | 0.18 | 0.55 | 0.90 |
| BIOSSES | 46.42 | 49.24 | 50.18 | 46.55 | 48.65 | 49.56 |
| SICK-R | 51.85 | 54.83 | 56.59 | 51.83 | 54.54 | 56.68 |
| STS12 | 26.79 | 29.32 | 30.65 | 27.07 | 29.50 | 31.42 |
| STS13 | 43.69 | 46.20 | 49.69 | 43.89 | 46.33 | 50.45 |
| STS14 | 36.85 | 39.21 | 41.86 | 36.87 | 39.25 | 42.38 |
| STS15 | 51.15 | 53.76 | 56.27 | 51.31 | 53.76 | 56.45 |
| STS16 | 48.44 | 50.22 | 52.40 | 48.45 | 50.54 | 52.02 |
| STS17 | 14.58 | 15.20 | 16.71 | 14.12 | 15.07 | 16.64 |
| STS22 | 33.20 | 34.21 | 34.74 | 33.81 | 34.34 | 35.20 |
| STSBenchmark | 37.81 | 41.20 | 44.44 | 37.72 | 41.11 | 44.51 |
| SummEval | 30.68 | 31.01 | 30.36 | 30.56 | 30.37 | 30.41 |

Table 13: Performance on all 56 MTEB datasets obtained on DeBERTa-base.

| | AdamW/3 | AdamW/5 | AdamW/10 | AdaTAMW/3 | AdaTAMW/5 | AdaTAMW/10 |
|---|---|---|---|---|---|---|
| AmazonCounterfactualClassification | 68.30 | 68.96 | 70.64 | 69.80 | 67.98 | 69.23 |
| AmazonPolarityClassification | 57.63 | 58.68 | 62.02 | 60.51 | 61.09 | 64.00 |
| AmazonReviewsClassification | 26.95 | 27.46 | 29.49 | 28.70 | 28.46 | 29.94 |
| Banking77Classification | 34.89 | 36.55 | 45.32 | 43.52 | 36.23 | 43.73 |
| EmotionClassification | 20.55 | 20.95 | 23.80 | 22.20 | 22.65 | 25.07 |
| ImdbClassification | 55.85 | 56.93 | 60.45 | 59.49 | 59.61 | 62.07 |
| MTOPDomainClassification | 49.99 | 50.56 | 55.26 | 52.74 | 47.11 | 53.55 |
| MTOPIntentClassification | 38.40 | 38.70 | 41.69 | 39.51 | 32.26 | 37.98 |
| MassiveIntentClassification | 22.93 | 22.17 | 24.10 | 23.32 | 19.29 | 22.66 |
| MassiveScenarioClassification | 25.17 | 25.24 | 27.90 | 26.03 | 24.65 | 27.54 |
| ToxicConversationsClassification | 58.13 | 58.67 | 62.96 | 62.53 | 60.96 | 62.62 |
| TweetSentimentExtractionClassification | 43.02 | 43.53 | 47.43 | 45.33 | 45.24 | 47.82 |
| ArxivClusteringP2P | 14.79 | 16.24 | 21.34 | 20.58 | 19.74 | 19.53 |
| ArxivClusteringS2S | 11.23 | 11.64 | 14.06 | 12.75 | 12.78 | 14.18 |
| BiorxivClusteringP2P | 7.13 | 8.33 | 14.75 | 12.45 | 11.97 | 15.23 |
| BiorxivClusteringS2S | 6.27 | 6.47 | 9.40 | 7.95 | 7.61 | 9.83 |
| MedrxivClusteringP2P | 13.91 | 14.76 | 18.62 | 17.15 | 16.65 | 18.55 |
| MedrxivClusteringS2S | 14.66 | 15.04 | 17.49 | 16.72 | 15.92 | 17.31 |
| RedditClustering | 10.31 | 10.58 | 14.61 | 13.25 | 12.33 | 15.42 |
| RedditClusteringP2P | 17.92 | 19.73 | 27.93 | 25.42 | 23.89 | 28.98 |
| StackExchangeClustering | 14.12 | 15.10 | 24.46 | 21.45 | 19.41 | 24.95 |
| StackExchangeClusteringP2P | 23.09 | 23.29 | 24.71 | 24.56 | 25.02 | 24.91 |
| TwentyNewsgroupsClustering | 8.72 | 9.11 | 13.04 | 11.73 | 11.53 | 13.04 |
| SprintDuplicateQuestions | 15.10 | 14.68 | 20.68 | 17.44 | 15.77 | 18.34 |
| TwitterSemEval2015 | 40.94 | 42.19 | 49.09 | 45.84 | 42.10 | 45.24 |
| TwitterURLCorpus | 58.85 | 58.74 | 64.98 | 60.50 | 56.99 | 61.66 |
| AskUbuntuDupQuestions | 43.00 | 42.49 | 44.44 | 43.29 | 43.21 | 43.89 |
| MindSmallReranking | 25.55 | 25.54 | 27.15 | 26.71 | 26.47 | 26.84 |
| SciDocsRR | 38.33 | 38.85 | 44.25 | 41.64 | 40.77 | 43.96 |
| StackOverflowDupQuestions | 29.51 | 29.47 | 31.44 | 30.02 | 28.81 | 30.51 |
| ArguAna | 2.82 | 3.56 | 9.89 | 7.38 | 7.41 | 11.42 |
| CQADupstackTexRetrieval | 0.08 | 0.09 | 0.39 | 0.32 | 0.34 | 0.68 |
| ClimateFEVER | 0.04 | 0.04 | 0.15 | 0.15 | 0.26 | 0.53 |
| DBPedia | 0.00 | 0.00 | 0.07 | 0.04 | 0.12 | 0.25 |
| FEVER | 0.01 | 0.01 | 0.11 | 0.12 | 0.20 | 0.31 |
| FiQA2018 | 0.04 | 0.07 | 0.37 | 0.16 | 0.26 | 0.50 |
| HotpotQA | 0.06 | 0.10 | 0.53 | 0.42 | 0.69 | 1.28 |
| MSMARCO | 0.03 | 0.04 | 0.09 | 0.08 | 0.12 | 0.13 |
| NFCorpus | 1.56 | 1.38 | 1.30 | 1.35 | 1.45 | 1.57 |
| NQ | 0.00 | 0.00 | 0.04 | 0.05 | 0.08 | 0.09 |
| QuoraRetrieval | 24.23 | 26.18 | 36.19 | 33.26 | 30.28 | 37.09 |
| SCIDOCS | 0.18 | 0.14 | 0.22 | 0.19 | 0.20 | 0.28 |
| SciFact | 0.38 | 0.45 | 1.01 | 0.95 | 0.81 | 2.58 |
| TRECCOVID | 3.84 | 4.12 | 6.46 | 5.93 | 4.28 | 6.21 |
| Touche2020 | 0.00 | 0.00 | 0.14 | 0.06 | 0.12 | 0.31 |
| BIOSSES | 45.61 | 46.15 | 44.77 | 43.29 | 34.63 | 44.84 |
| SICK-R | 46.29 | 45.12 | 51.62 | 47.51 | 44.42 | 50.14 |
| STS12 | 4.81 | 6.23 | 17.43 | 12.31 | 15.59 | 21.20 |
| STS13 | 29.36 | 30.41 | 40.42 | 34.59 | 32.32 | 37.90 |
| STS14 | 21.94 | 22.63 | 30.55 | 25.19 | 23.41 | 28.69 |
| STS15 | 37.17 | 35.34 | 47.61 | 39.55 | 40.42 | 44.24 |
| STS16 | 39.44 | 39.50 | 45.69 | 42.40 | 38.17 | 43.40 |
| STS17 | 21.97 | 20.99 | 20.99 | 20.05 | 17.87 | 20.32 |
| STS22 | 24.92 | 26.02 | 33.08 | 30.95 | 28.65 | 31.70 |
| STSBenchmark | 25.03 | 25.49 | 33.59 | 27.90 | 25.25 | 33.56 |
| SummEval | 30.48 | 30.46 | 30.51 | 30.32 | 29.16 | 30.29 |

Table 14: Performance on all 56 MTEB datasets obtained on DeBERTa-large.

| | AdamW/3 | AdamW/5 | AdamW/10 | AdaTAMW/3 | AdaTAMW/5 | AdaTAMW/10 |
|---|---|---|---|---|---|---|
| AmazonCounterfactualClassification | 69.49 | 69.31 | 69.16 | 69.25 | 69.11 | 68.98 |
| AmazonPolarityClassification | 65.74 | 65.46 | 65.44 | 65.63 | 65.55 | 65.81 |
| AmazonReviewsClassification | 26.57 | 26.53 | 26.55 | 26.55 | 26.50 | 26.52 |
| Banking77Classification | 63.52 | 63.33 | 63.51 | 64.02 | 64.19 | 64.62 |
| EmotionClassification | 32.81 | 32.95 | 33.05 | 33.38 | 33.45 | 33.52 |
| ImdbClassification | 58.48 | 58.44 | 58.41 | 58.55 | 58.56 | 58.83 |
| MTOPDomainClassification | 56.18 | 56.03 | 55.71 | 56.29 | 56.32 | 56.40 |
| MTOPIntentClassification | 39.80 | 39.63 | 39.64 | 39.67 | 39.93 | 39.89 |
| MassiveIntentClassification | 23.71 | 23.76 | 22.71 | 22.49 | 24.02 | 22.99 |
| MassiveScenarioClassification | 27.69 | 27.51 | 27.44 | 27.66 | 27.71 | 27.63 |
| ToxicConversationsClassification | 62.17 | 62.14 | 62.21 | 62.30 | 62.08 | 62.09 |
| TweetSentimentExtractionClassification | 52.11 | 52.11 | 51.97 | 52.13 | 52.10 | 51.92 |
| ArxivClusteringP2P | 23.44 | 23.41 | 23.51 | 23.75 | 23.67 | 24.13 |
| ArxivClusteringS2S | 20.04 | 20.06 | 20.06 | 20.08 | 20.16 | 20.37 |
| BiorxivClusteringP2P | 16.64 | 16.44 | 16.40 | 16.83 | 16.84 | 16.94 |
| BiorxivClusteringS2S | 18.19 | 18.07 | 18.14 | 18.41 | 18.33 | 18.36 |
| MedrxivClusteringP2P | 19.84 | 19.70 | 19.62 | 19.82 | 19.72 | 19.81 |
| MedrxivClusteringS2S | 20.54 | 20.52 | 20.46 | 20.52 | 20.50 | 20.46 |
| RedditClustering | 17.90 | 18.13 | 18.28 | 18.65 | 18.70 | 19.20 |
| RedditClusteringP2P | 25.73 | 25.85 | 25.97 | 26.49 | 26.47 | 26.88 |
| StackExchangeClustering | 34.70 | 34.99 | 34.97 | 35.89 | 36.02 | 36.48 |
| StackExchangeClusteringP2P | 24.86 | 24.79 | 24.89 | 25.00 | 24.98 | 25.13 |
| TwentyNewsgroupsClustering | 16.38 | 16.83 | 16.84 | 17.06 | 17.31 | 17.59 |
| SprintDuplicateQuestions | 47.92 | 47.66 | 47.15 | 48.07 | 48.06 | 48.23 |
| TwitterSemEval2015 | 53.50 | 53.48 | 53.65 | 53.63 | 53.68 | 53.88 |
| TwitterURLCorpus | 69.84 | 69.80 | 70.04 | 70.58 | 70.49 | 71.11 |
| AskUbuntuDupQuestions | 45.95 | 46.01 | 46.12 | 46.28 | 46.58 | 46.62 |
| MindSmallReranking | 27.71 | 27.68 | 27.70 | 27.73 | 27.69 | 27.77 |
| SciDocsRR | 52.86 | 52.87 | 52.83 | 53.40 | 53.42 | 53.52 |
| StackOverflowDupQuestions | 34.33 | 34.36 | 34.33 | 34.55 | 34.56 | 34.71 |
| ArguAna | 14.39 | 14.34 | 14.39 | 14.70 | 14.85 | 15.12 |
| CQADupstackTexRetrieval | 0.53 | 0.51 | 0.53 | 0.58 | 0.63 | 0.71 |
| ClimateFEVER | 0.26 | 0.26 | 0.24 | 0.28 | 0.26 | 0.27 |
| DBPedia | 0.40 | 0.35 | 0.41 | 0.42 | 0.48 | 0.70 |
| FEVER | 0.07 | 0.22 | 0.10 | 0.11 | 0.07 | 0.24 |
| FiQA2018 | 0.77 | 0.72 | 0.73 | 0.86 | 0.89 | 1.01 |
| HotpotQA | 1.14 | 1.09 | 1.07 | 1.17 | 1.17 | 1.51 |
| MSMARCO | 0.37 | 0.40 | 0.38 | 0.43 | 0.45 | 0.51 |
| NFCorpus | 1.47 | 1.47 | 1.48 | 1.52 | 1.54 | 1.64 |
| NQ | 0.29 | 0.27 | 0.28 | 0.29 | 0.26 | 0.34 |
| QuoraRetrieval | 55.52 | 55.54 | 55.67 | 56.44 | 56.52 | 57.00 |
| SCIDOCS | 0.41 | 0.40 | 0.40 | 0.43 | 0.42 | 0.45 |
| SciFact | 1.02 | 0.97 | 0.90 | 0.96 | 0.89 | 0.92 |
| TRECCOVID | 10.10 | 10.13 | 10.26 | 10.36 | 10.65 | 10.84 |
| Touche2020 | 0.07 | 0.07 | 0.06 | 0.09 | 0.13 | 0.20 |
| BIOSSES | 58.86 | 58.60 | 58.62 | 59.02 | 58.22 | 57.89 |
| SICK-R | 62.98 | 62.87 | 62.59 | 63.15 | 63.11 | 63.20 |
| STS12 | 33.84 | 34.14 | 34.17 | 35.40 | 35.67 | 36.82 |
| STS13 | 59.13 | 59.60 | 59.78 | 59.97 | 60.68 | 61.01 |
| STS14 | 47.29 | 47.61 | 48.46 | 49.70 | 49.91 | 51.35 |
| STS15 | 61.55 | 61.58 | 61.90 | 62.99 | 63.14 | 64.02 |
| STS16 | 62.84 | 63.13 | 63.17 | 62.83 | 63.64 | 63.18 |
| STS17 | 33.28 | 33.31 | 34.00 | 33.71 | 33.95 | 33.85 |
| STS22 | 22.91 | 22.84 | 22.76 | 22.76 | 23.10 | 23.26 |
| STSBenchmark | 54.87 | 54.91 | 55.02 | 55.66 | 56.02 | 57.20 |
| SummEval | 28.03 | 28.16 | 27.44 | 28.29 | 28.44 | 28.51 |

Table 15: Performance on all 56 MTEB datasets obtained on RoBERTa-base.

| | AdamW/3 | AdamW/5 | AdamW/10 | AdaTAMW/3 | AdaTAMW/5 | AdaTAMW/10 |
|---|---|---|---|---|---|---|
| AmazonCounterfactualClassification | 72.26 | 72.26 | 72.29 | 71.79 | 71.45 | 71.56 |
| AmazonPolarityClassification | 70.71 | 71.29 | 70.88 | 70.50 | 70.85 | 70.37 |
| AmazonReviewsClassification | 28.28 | 28.26 | 28.26 | 27.96 | 28.01 | 28.05 |
| Banking77Classification | 53.02 | 49.70 | 56.01 | 56.74 | 56.04 | 55.34 |
| EmotionClassification | 31.16 | 31.53 | 31.48 | 29.16 | 29.10 | 29.84 |
| ImdbClassification | 66.76 | 66.89 | 66.61 | 66.79 | 66.92 | 67.07 |
| MTOPDomainClassification | 62.52 | 61.11 | 60.38 | 60.71 | 60.38 | 59.86 |
| MTOPIntentClassification | 35.97 | 36.51 | 34.79 | 36.81 | 37.57 | 38.61 |
| MassiveIntentClassification | 24.01 | 21.85 | 21.99 | 24.89 | 23.15 | 23.15 |
| MassiveScenarioClassification | 30.51 | 31.41 | 31.30 | 30.36 | 31.76 | 31.18 |
| ToxicConversationsClassification | 66.41 | 66.76 | 66.56 | 65.64 | 65.68 | 65.31 |
| TweetSentimentExtractionClassification | 51.69 | 52.06 | 51.83 | 50.38 | 50.44 | 50.78 |
| ArxivClusteringP2P | 35.53 | 35.88 | 35.62 | 35.73 | 35.70 | 35.38 |
| ArxivClusteringS2S | 22.89 | 23.16 | 22.60 | 20.43 | 20.08 | 19.79 |
| BiorxivClusteringP2P | 31.56 | 31.57 | 31.69 | 31.58 | 31.63 | 31.37 |
| BiorxivClusteringS2S | 21.31 | 21.42 | 20.99 | 20.05 | 19.75 | 19.66 |
| MedrxivClusteringP2P | 27.20 | 27.25 | 27.24 | 27.06 | 27.29 | 27.38 |
| MedrxivClusteringS2S | 22.90 | 23.02 | 22.71 | 21.99 | 21.87 | 21.84 |
| RedditClustering | 25.38 | 26.41 | 25.73 | 21.14 | 21.11 | 21.33 |
| RedditClusteringP2P | 44.90 | 45.24 | 44.58 | 44.47 | 44.78 | 44.89 |
| StackExchangeClustering | 45.02 | 46.10 | 45.34 | 39.25 | 38.89 | 39.37 |
| StackExchangeClusteringP2P | 26.31 | 26.34 | 26.29 | 26.17 | 26.17 | 26.16 |
| TwentyNewsgroupsClustering | 22.50 | 22.60 | 22.25 | 15.39 | 15.71 | 15.64 |
| SprintDuplicateQuestions | 57.43 | 58.21 | 57.79 | 47.77 | 47.07 | 43.12 |
| TwitterSemEval2015 | 49.58 | 50.37 | 50.32 | 50.36 | 50.50 | 49.86 |
| TwitterURLCorpus | 69.08 | 69.63 | 69.35 | 66.71 | 67.40 | 67.56 |
| AskUbuntuDupQuestions | 47.54 | 47.62 | 47.36 | 47.16 | 47.55 | 47.22 |
| MindSmallReranking | 28.69 | 28.60 | 28.81 | 27.89 | 27.72 | 27.70 |
| SciDocsRR | 57.88 | 58.36 | 57.86 | 53.35 | 53.26 | 53.56 |
| StackOverflowDupQuestions | 34.50 | 34.59 | 34.38 | 34.74 | 35.12 | 34.39 |
| ArguAna | 26.35 | 26.68 | 26.59 | 26.95 | 27.58 | 28.30 |
| CQADupstackTexRetrieval | 1.78 | 2.02 | 1.83 | 2.87 | 2.92 | 2.47 |
| ClimateFEVER | 6.09 | 4.90 | 4.88 | 7.84 | 8.26 | 6.97 |
| DBPedia | 2.46 | 2.53 | 2.14 | 3.81 | 4.24 | 4.08 |
| FEVER | 3.40 | 2.09 | 2.41 | 7.76 | 5.87 | 4.46 |
| FiQA2018 | 2.74 | 3.28 | 3.13 | 4.41 | 4.26 | 4.44 |
| HotpotQA | 6.36 | 6.46 | 5.83 | 8.15 | 10.10 | 8.05 |
| MSMARCO | 1.86 | 1.82 | 1.65 | 1.92 | 2.23 | 2.16 |
| NFCorpus | 3.37 | 3.56 | 3.43 | 3.27 | 3.67 | 3.53 |
| NQ | 3.61 | 3.73 | 3.49 | 4.17 | 4.80 | 4.75 |
| QuoraRetrieval | 57.87 | 58.94 | 58.46 | 57.32 | 57.04 | 58.19 |
| SCIDOCS | 1.73 | 1.92 | 1.90 | 2.27 | 2.40 | 2.45 |
| SciFact | 14.09 | 15.50 | 14.82 | 19.00 | 18.92 | 17.34 |
| TRECCOVID | 15.39 | 15.73 | 14.60 | 17.65 | 17.08 | 16.57 |
| Touche2020 | 1.68 | 1.56 | 1.51 | 3.16 | 2.55 | 2.45 |
| BIOSSES | 57.46 | 58.08 | 57.91 | 56.01 | 55.86 | 52.38 |
| SICK-R | 58.12 | 57.90 | 58.14 | 53.95 | 54.43 | 53.28 |
| STS12 | 30.82 | 30.83 | 28.26 | 31.42 | 32.37 | 28.59 |
| STS13 | 53.15 | 54.49 | 52.35 | 51.01 | 52.27 | 50.47 |
| STS14 | 43.24 | 44.51 | 42.35 | 42.08 | 43.27 | 41.76 |
| STS15 | 54.42 | 56.06 | 54.74 | 53.15 | 54.52 | 54.94 |
| STS16 | 58.91 | 58.67 | 59.83 | 55.41 | 54.53 | 55.37 |
| STS17 | 28.40 | 27.01 | 26.63 | 16.20 | 17.53 | 16.19 |
| STS22 | 25.02 | 26.14 | 25.54 | 24.60 | 24.69 | 24.55 |
| STSBenchmark | 54.42 | 54.66 | 54.29 | 50.47 | 50.82 | 49.29 |
| SummEval | 29.43 | 29.71 | 29.59 | 29.18 | 29.37 | 29.01 |

Table 16: Performance on all 56 MTEB datasets obtained on RoBERTa-large.

