# OpenReview forum: "Torque-Aware Momentum"
_ICLR.cc/2025/Conference — Submitted to ICLR 2025_

### Official Review · Reviewer_fkEA · 2024-10-18

**Soundness:** 2
**Presentation:** 2
**Contribution:** 2
**Rating:** 5
**Confidence:** 4

**Summary:**

This paper identified the problem of momentum that large and misaligned gradients can lead to oscillations during training of DNNs. Thus, it proposed a novel momentum method called Torque-Aware Momentum (TAM), which introduces a damping factor based on the angle between the new gradients and previous momentum, stabilizing the update direction during training. The experiments demonstrate the effectiveness of TAM.

**Strengths:**

-As Momentum is a very important technique for optimization and deep learning, it is always interesting to see a novel and effective mometum method, such as TAM.

-The experiments cover both vision tasks and language tasks.

-The idea of TAM is elegant and reasonable.

**Weaknesses:**

-This paper lacks convergence analysis. As a optimization method, the convergence guarantee is expected. And many previous studies on momentum provided convergence analysis.

-While large oscillations are bad, oscillations are sometimes good for searching minima intuitively. This paper also failed to formally explain the generalization advantage of TAM over standard momentum. Generalization bound analysis can be helpful.

-The empirical improvements are marginal, especially for vision tasks. The experiments of language modeling are also not compressive.

-The work missed a lot recent relevant works on analyzing or upgrading momentum. For example, [1,2] theoretically studied various momentum methods; [3, 4] designed momentum variants.

Refs:

[1] Yan, Y., Yang, T., Li, Z., Lin, Q., & Yang, Y. A unified analysis of stochastic momentum methods for deep learning. IJCAI 2017.

[2] Liu, Y., Gao, Y., & Yin, W. An improved analysis of stochastic gradient descent with momentum. NeurIPS 2020.

[3] Xie, Z., Yuan, L., Zhu, Z., & Sugiyama, M. Positive-negative momentum: Manipulating stochastic gradient noise to improve generalization. ICML 2021.

[4] Cutkosky, A., & Mehta, H. Momentum improves normalized sgd. In International conference on machine learning. ICML 2021.

**Questions:**

Please see the weaknesses.

---

> ### Author Response · Authors · 2024-11-21
>
> We thank the reviewer for their evaluation and for recognizing the novelty and elegance of our proposed TAM method, as well as the breadth of experiments across vision and language tasks. We also appreciate the constructive suggestions, which we address in detail below.
>
> **Reviewer’s Comment: “This paper lacks convergence analysis.”**
>
> **Response**:  We understand the reviewer's concern about convergence proof. To address this, we propose to add an argument that leverages TAM's relationship with SGDM at late training times discussed in Section 3 (“Learning Rate Transfer”). The argument goes as follows:
> - As training progresses, TAM's cosine similarity term $ŝ_t$ stabilizes to a constant value $s^*$ (which we empirically observed to be approximately 0, see Appendix A.2.1).
> - Under this late-time behavior, TAM's update rule effectively reduces to SGDM with a modified learning rate: $ \eta^{\text {eff}}_\text{TAM} \approx \frac{1+{s}^*}{2(1-\beta)}\eta $.
> - This equivalence means that in the neighborhood of optima, where $\hat{s}_t$ has stabilized, TAM inherits the well-established convergence guarantees of SGDM  [1,2].
> - The damping factor $(1 + \hat{s}_t)/2$ remains bounded, ensuring the effective learning rate stays within a controlled range throughout training.
>
> This theoretical connection to SGDM, combined with our empirical evidence of $s^*$ stabilizing to $0$, ensures TAM's convergence while maintaining its enhanced exploration capabilities during early training. We've added this explanation in Section 3 in the revision.
>
> -------------
>
> **Reviewer’s Comment: “While large oscillations are bad, oscillations are sometimes good for searching minima intuitively.”**
>
> **Response**:  We agree that oscillations can sometimes be beneficial for exploring the loss landscape. As shown in Figure 6, TAM does not eliminate oscillations entirely. Instead, it defers abrupt sharpening events caused by large oscillations, promoting smoother exploration. Additionally, our LMC analysis highlights how TAM enhances training stability, which we argue translates to better generalization performance. To strengthen this point, we have also added more gradient norm plots to the Appendix A.2.6, as well as an analysis of $\hat{s}_t$ (cosine similarity) in Appendix A.2.1, which provides a more direct measure of oscillatory behavior. We hope these additions address your concerns.
>
> -------------
>
> **Reviewer’s Comment: “The empirical improvements are marginal, especially for vision tasks.”**
>
> **Response**:  We would like to emphasize that the primary goal of TAM is to enhance momentum-based optimizers by stabilizing gradient directions and improving exploration of the loss landscape. While we may expect TAM to match the performance of standard momentum methods in conventional supervised learning settings, its strengths become evident in more complex scenarios involving misaligned gradients or task adaptation:
>
> - **Fine-tuning**: Fine-tuning on new datasets while retaining previous knowledge is challenging. TAM outperforms other optimizers in avoiding sharp minima and improving exploration, as shown in Fig. 3 and Fig. 4 across different models and compute budgets. Notably, AdaTAM achieves better performance on the Wikitext dataset and generalizes well on most MTEB datasets without overfitting.
> - **Online learning**: In non-stationary environments, where maintaining plasticity to adapt to new tasks is crucial, TAM demonstrates superior adaptability across task boundaries (Fig. 5).
> - **Continual learning**: Following *Reviewer fh59*’s suggestion, we also added new continual learning experiments in Appendix A.2.3, comparing SGDM and TAM on the CLEAR dataset (with 10 sequential tasks). The results confirm that TAM outperforms SGDM in settings that require both stability and plasticity.
>
> Therefore, all these experiments highlight the advantages of TAM in stabilizing momentum updates and enhancing exploration in such settings.
>
>
> -------------
>
>
> **Reviewer’s Comment: “The work missed a lot recent relevant works on analyzing or upgrading momentum. For example, [1,2] theoretically studied various momentum methods; [3, 4] designed momentum variants.”**
>
> **Response**:  We appreciate the reviewer bringing these works to our attention. We have added the suggested references to the manuscript and updated the related work section to acknowledge their contributions.
>
> -------------
>
> [1] Yan, Y., Yang, T., Li, Z., Lin, Q., & Yang, Y. A unified analysis of stochastic momentum methods for deep learning. IJCAI 2017.
>
> [2] Liu, Y., Gao, Y., & Yin, W. An improved analysis of stochastic gradient descent with momentum. NeurIPS 2020.
>
> -------------
>
> **We thank the reviewer for their constructive feedback, which has helped improve our work. We believe our responses address the concerns raised, and we would appreciate it if the updated score reflects this. Please let us know if further clarifications are needed.**

---

> > ### Comment · Reviewer_fkEA · 2024-11-25
> > **Thanks for the responses.**
> >
> > Thanks for the responses. They addressed some of my concerns.
> >
> > However, some main weaknesses still exist, theoretically and empirically.
> >
> > 1) The proposed optimization still lacks formal convergence theoretical guarantee.
> >
> > 2) The empirical improvement are indeed marginal.
> >
> >
> > According to the responses, I consider this work is a borderline work for ICLR. So I raise the score from 3 to 5.

---

> > > ### Author Response · Authors · 2024-11-25
> > >
> > > Thank you for acknowledging our rebuttal and for raising your score. However, we believe our rebuttal already addresses the two points raised:
> > >
> > > 1. Under empirically validated assumptions, we demonstrated that TAM updates asymptotically reduce to SGDM updates, ensuring that TAM inherits the convergence guarantees of SGDM. While we could formalize this into a “theorem-proof” structure in a revised version, we are uncertain if this would add significant clarity. We would appreciate your opinion on whether such framing would meaningfully enhance the paper.
> > >
> > > 2. We respectfully disagree regarding the empirical improvements. As shown in our submission and the new experiments included in the rebuttal, TAM delivers substantial gains in contexts where misaligned gradients have a pronounced impact (e.g., fine-tuning, online learning, continual learning), while matching the performance of standard momentum methods in conventional setups. Therefore TAM successfully fulfills its primary objective: enhancing robustness to gradient misalignment and improving exploration of the loss landscape.
> > >
> > > We hope the reviewer will reconsider these points in evaluating the contributions and significance of our work.

---

### Official Review · Reviewer_Di8u · 2024-11-03

**Soundness:** 3
**Presentation:** 3
**Contribution:** 3
**Rating:** 3
**Confidence:** 4

**Summary:**

1. Summary	This paper is based on the idea that rapid changes in gradients (torque) negatively impact the performance of deep learning models. It attempts to improve the generalization performance of optimizers that utilize momentum (e.g., SGDM or Adam) by applying a damping effect during momentum updates, which is based on the angle between the gradient and momentum. The author devised a variation of SGDM called TAM and a variation of Adam(W), referred to as AdaTAM(W), and trained deep learning models on various datasets and tasks. Additionally, the author demonstrated the performance of the TAM approach in online learning. In the ResNet18 and CIFAR10 environments, the author showed that TAM exhibits less oscillation compared to SGDM.

**Strengths:**

1. A damping effect was applied based on the cosine similarity, which intuitively illustrates the relationship between gradients and momentum.
2. The experiments conducted on the Language Model (LM) side involved a diverse range of datasets.
3. The small change in gradient norm on the CIFAR10 dataset suggests that TAM has some effect in mitigating oscillations.

**Weaknesses:**

1. Throughout the paper, TAM and AdaTAM(W) are evaluated using a limited set of models, raising questions about whether the TAM approach works only with specific models. For instance, in the image domain, only ResNet-based models were used, while in the language model (LM) domain, only BERT-based models were employed. It is recommended to use other models in the image domain, such as MobileNet and ViT, and in the language domain, models like GPT, T5, and LLaMA.
2. In Table 1, the performance of TAM on the ImageNet dataset is quite marginal, with only a 0.1%-point difference when comparing SGDM and TAM (77.0 vs. 77.1). This is also the case when comparing Adam and AdaTAM (74.4 vs. 74.5. Since the aim of the method is to improve generalization performance, one would expect a more significant performance difference, especially on large datasets like ImageNet.
3. While it is impressive that TAM exhibits less oscillation compared to SGDM in Figure 6, the relationship between AdaTAM and Adam is not shown. To convincingly demonstrate that the proposed damping is indeed effective, it would be beneficial to include graphs obtained from AdaTAM as well.
4. In Figure 4, the threshold for similarity is set at 1%-point, which is a large value and could lead to the conclusion that they are not similar. It is recommended to lower the threshold for similarity to around 0.2%-point.
5. There is no convergence proof for TAM and AdaTAM. Since optimizers are techniques used in artificial intelligence training, it is essential to demonstrate that the models converge when these methods are applied. It is recommended to at least provide mathematical proof showing that the proposed optimizers converge in a convex environment. The convergence of Adam has already been proven, and several other optimizer papers have provided similar proofs (e.g., AMSGrad, AdaMax, SGDM).

**Questions:**

1. Why does TAM diverge at epoch 0 in the middle figure of Figure 6?
2. In the Methods section, gamma is fixed, but what is the effect of varying gamma on performance? An ablation study on this would be helpful. It would be useful to understand how performance changes with hyperparameters, as seen in Adam.
3. In the case of Adam, an exponential moving average is used to update momentum; why is this not done in AdaTAM?
4. Only the fine-tuning results for the Language Model (LM) are presented, but there are no results for training from scratch. I am curious about how performance would change if the TAM approach were used from the pre-training stage of the LM.
5. Figure 6 presents results for CIFAR10, but it is difficult to validate the performance of TAM using CIFAR10 alone. Are there graphs for more complex datasets, such as CIFAR100, which has a greater number of classes?

---

> ### Author Response · Authors · 2024-11-21
> **Response 1/3**
>
> Thank you for your detailed and thoughtful review. We appreciate your recognition of the strengths of our work, particularly in applying the damping effect based on cosine similarity, the diverse set of experiments on language models, and the observed impact of TAM on gradient stability. We also appreciate the suggestions and have provided our responses to the concerns below:
>
> **Reviewer’s Comment: “TAM and AdaTAM(W) are evaluated using a limited set of models, ... It is recommended to use other models in the image domain ... how performance would change if the TAM approach were used from the pre-training stage of the LM.”**
>
>
> **Response**: Thank you for your suggestion. In response, we have expanded our experiments to include additional models. Specifically, we have evaluated TAM with MobileNet on CIFAR-10/100 and ViT fine-tuning (pre-trained on ImageNet-1k). We perform the learning rate grid search to obtain the best performing setup. Details of these experiments are included in the revised version of the paper in Appendix A.2.2. We also provide the results below:
>
>
> ### CIFAR10 + MobileNet:
> | **Method**          | **Accuracy**      | **Best LR**      |
> |----------------------|-------------------|-------------|
> | SGD                  | 93.7 ± 0.2       | 0.1         |
> | SGDM                 | 93.9 ± 0.1       | 0.01        |
> | TAM (ours)           | 93.9 ± 0.2       | 0.02        |
> | Adam                 | 92.7 ± 0.1       | 0.01        |
> | AngularGrad          | 91.7 ± 0.1       | 0.001       |
> | AdaTAM (ours)        | 93.1 ± 0.1       | 0.0001      |
>
>
> ### CIFAR100 + MobileNet:
> | **Method**          | **Accuracy**      | **Best LR**      |
> |----------------------|-------------------|-------------|
> | SGD                  | 72.8 ± 0.1       | 0.1         |
> | SGDM                 | 72.8 ± 0.3       | 0.01        |
> | TAM (ours)           | 72.8 ± 0.1       | 0.02        |
> | Adam                 | 69.8 ± 0.1       | 0.001       |
> | AngularGrad          | 66.4 ± 0.1       | 0.001       |
> | AdaTAM (ours)        | 70.7 ± 0.3       | 0.0001      |
>
>
> ### CIFAR10 + ViT:
> | **Method**          | **Accuracy**      | **Best LR** |
> |----------------------|-------------------|-------------|
> | SGD                  | 97.1 ± 0.1       | 0.1         |
> | SGDM                 | 97.7 ± 0.1       | 0.1         |
> | TAM (ours)           | 97.7 ± 0.1       | 0.2         |
>
>
> ### CIFAR100 + ViT:
> | **Method**          | **Accuracy**      | **Best LR**      |
> |----------------------|-------------------|-------------|
> | SGD                  | 74.4 ± 0.6       | 0.03        |
> | SGDM                 | 85.3 ± 0.2       | 0.1         |
> | TAM (ours)           | 86.2 ± 0.2       | 0.2         |
>
> Overall, we observe that TAM and AdaTAM either match the performance or outperform the baselines similar to results in Table 1.
>
> We also would like to inform the reviewer that we are also currently running experiments on training GPT, which we will update the paper with when they are finished.
>
> -------------
>
> **Reviewer’s Comment: “In Table 1, the performance of TAM on the ImageNet dataset is quite margina ...”**
>
> **Response**:  We would like to emphasize that the primary goal of TAM is to enhance momentum-based optimizers by stabilizing gradient directions and improving exploration of the loss landscape. While we may expect TAM to match the performance of standard momentum methods in conventional settings, its strengths become evident in more complex scenarios involving misaligned gradients or task adaptation:
>
> - **Fine-tuning**: Fine-tuning on new datasets while retaining previous knowledge is challenging. TAM outperforms other optimizers in avoiding sharp minima and improving exploration, as shown in Fig. 3 and Fig. 4 across different models and compute budgets. Notably, AdaTAM achieves better performance on the Wikitext dataset and generalizes well on most MTEB datasets without overfitting.
> - **Online learning**: In non-stationary environments, where maintaining plasticity to adapt to new tasks is crucial, TAM demonstrates superior adaptability across task boundaries (Fig. 5).
> - **Continual learning**: Following *Reviewer fh59*’s suggestion, we also added new continual learning experiments in Appendix A.2.3, comparing SGDM and TAM on the CLEAR dataset (with 10 sequential tasks). The results confirm that TAM outperforms SGDM in settings that require both stability and plasticity.
>
> Therefore, all these experiments highlight the advantages of TAM in stabilizing momentum updates and improving exploration in such settings.
>
>
> -------------

---

> ### Author Response · Authors · 2024-11-21
> **Response 2/3**
>
> **Reviewer’s Comment: “ While it is impressive that TAM exhibits less oscillation compared to SGDM in Figure 6, the relationship between AdaTAM and Adam is not shown ... it would be beneficial to include graphs obtained from AdaTAM as well.”**
>
>
> **Response**: Thank you for your suggestion. We have added a plot in the Appendix A.2.6 comparing AdaTAM and Adam (Figure 9-right), showing the evolution of gradient norm (Grad Norm) using the same setup as in Fig. 6, with a fixed learning rate of 0.001 for both adaptive optimizers. The results demonstrate that while both optimizers experience a high gradient norm in the first epoch, AdaTAM consistently maintains a lower gradient norm as training progresses, indicating that the damping effect is effectively controlling large gradients in AdaTAM.
>
> -------------
>
> **Reviewer’s Comment: “the threshold for similarity is set at 1%-point ... It is recommended to lower the threshold for similarity to around 0.2%-point.”**
>
>
> **Response**:  Thank you for your recommendation. We have updated the results in Figure 4 to use a 0.2%-point similarity threshold and moved the previous 1%-point results to the Appendix A.2.7. With this change, we observe that, except for RoBERTa-large and BERT-base fine-tuned for 10 epochs, AdaTAMW generally matches or outperforms AdamW in most settings.
>
> -------------
>
> **Reviewer’s Comment: “There is no convergence proof for TAM and AdaTAM. ”**
>
> **Response**: We understand the reviewer's concern about convergence proof. To address this, we propose to add an argument that leverages TAM's relationship with SGDM at late training times discussed in Section 3 (“Learning Rate Transfer”). The argument goes as follows:
> - As training progresses, TAM's cosine similarity term $ŝ_t$ stabilizes to a constant value $s^*$ (which we empirically observed to be approximately 0, see Appendix A.2.1).
> - Under this late-time behavior, TAM's update rule effectively reduces to SGDM with a modified learning rate: $ \eta^{\text {eff}}_\text{TAM} \approx \frac{1+{s}^*}{2(1-\beta)}\eta $.
> - This equivalence means that in the neighborhood of optima, where $\hat{s}_t$ has stabilized, TAM inherits the well-established convergence guarantees of SGDM [1,2].
> - The damping factor $(1 + \hat{s}_t)/2$ remains bounded, ensuring the effective learning rate stays within a controlled range throughout training.
>
> This theoretical connection to SGDM, combined with our empirical evidence of $s^*$ stabilizing to $0$, ensures TAM's convergence while maintaining its enhanced exploration capabilities during early training. We've added this explanation in Section 3 in the revision.
>
> -------------
>
> **Reviewer’s Comment: “Why does TAM diverge at epoch 0 in the middle figure of Figure 6?”**
>
> **Response**: We would like to clarify that TAM does not diverge at epoch 0 in the middle figure of Fig 6. The gradient norm starts at a higher value for all methods, including TAM, and decreases quickly after the first epoch before increasing again in later stages due to abrupt sharpening. To address the confusion, we have updated Figure 6 to improve clarity.
>
>
> -------------
>
> **Reviewer’s Comment: “It would be useful to understand how performance changes with hyperparameters, as seen in Adam.”**
>
>
> **Response**: Thank you for this suggestion. We conducted an ablation study on ResNet18, varying gamma while keeping all other hyperparameters fixed for CIFAR10 and CIFAR100. Our experiments indicate that varying gamma has minimal impact on the overall behavior of the optimization trajectories. The results, included in the Appendix A.2.4, demonstrate that even with changes in gamma, TAM consistently outperforms other baselines. The results are as follows:
>
> ### CIFAR10
>
> | $\gamma$ | Accuracy          |
> |----------|----------------------|
> | 0.99     | 93.9 +/- 0.1         |
> | 0.9      | 94.2 +/- 0.2 (reported in Table 1) |
> | 0.8      | 94.1 +/- 0.2         |
> | 0.5      | 93.9 +/- 0.1         |
> | 0        | 94.1 +/- 0.1         |
>
> ### CIFAR100
>
> | $\gamma$ | Accuracy          |
> |----------|----------------------|
> | 0.99     | 74.0 +/- 0.1         |
> | 0.9      | 73.8 +/- 0.1 (reported in Table 1) |
> | 0.8      | 74.1 +/- 0.3         |
> | 0.5      | 73.7 +/- 0.4         |
> | 0        | 74.1 +/- 0.3         |
>
>
> -------------

---

> > ### Author Response · Authors · 2024-11-21
> > **Response 3/3**
> >
> > **Reviewer’s Comment: “In the case of Adam, an exponential moving average is used to update momentum; why is this not done in AdaTAM?”**
> >
> > **Response**: Thank you for pointing this out. We empirically observed that incorporating an exponential moving average into AdaTAM (referred to as AdaTAM2) had minimal impact on performance and, in some cases, slightly degraded it. For example, on CIFAR100, AdaTAM2 yielded slightly lower test accuracy compared to the default AdaTAM. We have this comparison in Appendix A.2.5.
> >
> > ### Performance Comparison
> >
> > | Setup                  | AdaTAM2           | (Best LR)   | AdaTAM (reported in Table 1)    |
> > |------------------------|-------------------|-----------|----------------------|
> > | CIFAR10 + Resnet 18     | 93.3 +/- 0.1      | (0.0001)  | 93.3 +/- 0.3         |
> > | CIFAR10 + Resnet 34     | 93.6 +/- 0.2      | (0.0001)  | 93.3 +/- 0.1         |
> > | CIFAR100 + Resnet 18    | 71.9 +/- 0.2      | (0.001)   | 72.7 +/- 0.3         |
> > | CIFAR100 + Resnet 34    | 72.6 +/- 0.1      | (0.0001)  | 72.9 +/- 0.1         |
> >
> >
> > -------------
> >
> >
> > **Reviewer’s Comment: “Figure 6 presents results for CIFAR10, but it is difficult to validate the performance of TAM using CIFAR10 alone.”**
> >
> > **Response**: Thank you for the suggestion. We have added gradient norm plots on SVHN and CIFAR100 in the Appendix A.2.6. On SVHN, we observe a similar pattern as with CIFAR10: while both optimizers experience abrupt jumps (likely due to oscillations in sharp minima), TAM defers this oscillatory behavior and explores the loss landscape for more epochs. On CIFAR100, we observe that TAM completely avoids abrupt jumps in gradient norm within the first 100 epochs. Moreover, SGDM with $sw=25$ also demonstrates controlled gradient norms, indicating improved training stability.
> >
> > -------------
> >
> >
> > [1] Yan, Y., Yang, T., Li, Z., Lin, Q., & Yang, Y. A unified analysis of stochastic momentum methods for deep learning. IJCAI 2017.
> >
> > [2] Liu, Y., Gao, Y., & Yin, W. An improved analysis of stochastic gradient descent with momentum. NeurIPS 2020.
> >
> > -------------
> >
> > **We thank the reviewer for their constructive feedback, which has helped improve our work. We believe our responses address the concerns raised, and we would appreciate it if the updated score reflects this. Please let us know if further clarifications are needed.**

---

> > > ### Author Response · Authors · 2024-11-25
> > >
> > > Dear Reviewer Di8u, we hope that you've had a chance to read our responses and clarification. As the discussion period is ending soon, we would greatly appreciate it if you could confirm whether our updates have addressed your concerns and, if possible, support our work by considering an increased score.

---

> > > > ### Author Response · Authors · 2024-11-28
> > > >
> > > > Dear Reviewer Di8u, we hope that you've had a chance to read our responses and clarification. As a gentle reminder, we would greatly appreciate it if you could confirm whether our updates have addressed your concerns and, if possible, support our work by considering an increased score.

---

### Official Review · Reviewer_fh59 · 2024-11-04

**Soundness:** 3
**Presentation:** 3
**Contribution:** 3
**Rating:** 6
**Confidence:** 4

**Summary:**

This paper proposes Torque-Aware Momentum (TAM), an improvement to traditional momentum-based optimizers, designed to handle large, misaligned gradients that may lead to oscillations and hinder convergence in DL. Through incorporating a damping factor adjusting the momentum update based on the alignment angle between current gradients and prior momentum, TAM aims to make the training stable, improve generalization ability, and increase exploration of the loss landscape. The proposed approach is evaluated over a range of tasks which include image classification, LLM fine-tuning, and online learning, indicating consistent improvements over standard momentum-based methods such as  SGDM.

**Strengths:**

1. The paper proposes an innovative concept via connecting damping mechanics to momentum updates, which adds an insightful dimension to the optimization community in ML.
2. The paper is equipped with detailed empirical analysis, which includes the comparisons over a variety of benchmarks and model types, highlighting the robustness and wide applicability of TAM proposed.
3. The concept of incorporating a damping factor contigent on gradient alignment is well-explained, and the pseudo-code presented contributes to clarifying implementation.

**Weaknesses:**

1. Despite the fact that TAM is computationally efficient, the requirement to compute the cosine similarity between gradients and momentum might introduce non-trivial implementation complexity in certain training frameworks.
2. While comparisons with the standard optimizers and a few existing approachs are robust, additional benchmarks against more recent optimizers concentrating on gradient stability would strengthen the claims furthermore.
3. The paper illustrates TAM's effectiveness on various tasks, but its performance in more demanding and complex training scenarios, such as very large LLMs (e.g., 70B), could be further explored.

**Questions:**

Would TAM still perform well in non-stationary environments, such as continual learning, where the gradient directions frequently change or shift?

---

> ### Author Response · Authors · 2024-11-21
>
> We’d like to thank the reviewer for the overall positive evaluation of our paper, specifically the appreciation of the novelty,  robustness and wide applicability of our method, and the clarity of our exposition. We appreciate the suggestions and have provided our responses below:
>
> **Reviewer’s Comment: “Despite the fact that TAM is computationally efficient, the requirement to compute the cosine similarity between gradients and momentum might introduce non-trivial implementation complexity in certain training frameworks.”**
>
>  We would like to clarify that TAM only requires the computation of the cosine similarity between two precomputed vectors (gradient and momentum), which involves a simple normalized dot product. This operation is computationally efficient and induces negligible overhead in modern training frameworks. Furthermore, TAM does not require any additional memory beyond storing the gradient and momentum vectors, which are already maintained in standard optimizers like SGDM.
>
> Please note that we also provide a runtime time comparison in training BERT models in Appendix A.2.8.
>
> -------------
>
> **Reviewer’s Comment: “While comparisons with the standard optimizers and a few existing appoachs are robust, additional benchmarks against more recent optimizers concentrating on gradient stability would strengthen the claims furthermore.”**
>
> Thank you for the suggestion. We have compared TAM with AngularGrad, the closest baseline targeting gradient stability. If you have specific recommendations for additional recent optimizers to benchmark against, we would be happy to consider them in future experiments.
>
>
> -------------
>
> **Reviewer’s Comment: “The paper illustrates TAM's effectiveness on various tasks, but its performance in more demanding and complex training scenarios, such as very large LLMs (e.g., 70B), could be further explored.”**
>
> Thank you for the suggestion. While running experiments on very large-scale models like 70B LLMs is well beyond our current computational resources, we have included evaluations across a range of model sizes, including BERT variants, to demonstrate TAM's scalability and effectiveness. We believe these experiments provide sufficient evidence of TAM's robustness and general applicability across diverse scenarios.
>
>
> -------------
>
> **Reviewer’s Comment: “Would TAM still perform well in non-stationary environments, such as continual learning, where the gradient directions frequently change or shift?”**
>
> Yes, TAM is particularly well-suited for non-stationary settings, as exploring the loss surface and preserving previously learned features are critical for maintaining a balance between plasticity and stability in continual learning. In Fig. 5, we demonstrate TAM's superior ability to adapt to new tasks across different levels of difficulty (task boundaries).
>
> To further validate this, in response to the reviewer’s comment, we conducted continual learning experiments using a pre-trained ResNet-50 on the CLEAR dataset, which consists of 10 sequential tasks [1]. We compare TAM with SGDM across various learning rates on top of two continual learning setups: (i) Naive and (ii) Learning without forgetting (LwF) [2] which is a well-known continual learning method. We summarize the results obtained using the best-performing setup below:
>
> | **Methods** | **Optimizers** | **Average final accuracy** |
> |-------------|----------------|-------------|
> | **Naive**   | SGDM           | 90.63%       |
> |             | TAM (ours)     | 92.70%       |
> | **LwF**     | SGDM           | 93.76%       |
> |             | TAM (ours)     | 95.14%       |
>
>
> Detailed results are provided in Appendix A.2.3 of the revision. These results again show the effectiveness of TAM in a challenging setting where a model is required to maintain both stability and plasticity.
>
> -------------
>
> [1] Xiaohui Zhang, Jiangyan Yi, Jianhua Tao, Chenglong Wang, and Chu Yuan Zhang. Do you remember? overcoming catastrophic forgetting for fake audio detection. In International Conference on Machine Learning, pp. 41819–41831. PMLR, 2023.
>
> [2] Zhizhong Li and Derek Hoiem. Learning without forgetting. IEEE transactions on pattern analysis and machine intelligence, 40(12):2935–2947, 2017.
>
> -------------
>
> **We appreciate the review and its valuable feedback. We hope our responses address the concerns raised. If so, we kindly request this be reflected in the score. Please feel free to reach out with any further questions, and we will respond promptly.**

---

> > ### Author Response · Authors · 2024-11-25
> >
> > Dear Reviewer fh59, we hope that you've had a chance to read our responses and clarification. As the discussion period is ending soon, we would greatly appreciate it if you could confirm whether our updates have addressed your concerns and, if possible, support our work by considering an increased score.

---

> > > ### Author Response · Authors · 2024-11-28
> > >
> > > Dear Reviewer fh59, we hope that you've had a chance to read our responses and clarification. As a gentle reminder, we would greatly appreciate it if you could confirm whether our updates have addressed your concerns and, if possible, support our work by considering an increased score.

---

> ### Comment · Reviewer_fh59 · 2024-11-28
> **Thanks for the response**
>
> Thanks for the authors' response which solves my concerns to a considerable extent. After consideration, I decide to maintain my score.

---

### Author Response · Authors · 2024-11-21

We thank all the reviewers for their questions, comments and suggestions. We are glad that the reviewers found our idea innovative (Reviewer fh59,fkEA), intuitive (Reviewer di8u) and reasonable (Reviewer fkEA). We also appreciate that the reviewers acknowledge the detailed empirical analysis (Reviewer di8u,fkEA), demonstrating the robustness and wide applicability of TAM (Reviewer fh59), presented in our paper.


We have responded to specific concerns and list the changes we made to the revised manuscript here:
- Added missing references in Section 2 and Section 3.
- Added an explanation on how TAM’s convergence is related to SGDM by expanding on learning rate transfer analysis.
- Updated Figure 4 based on Reviewer di8u’s suggestion on lowering the threshold for similarity to around 0.2%-point. Moved the older Figure to Appendix A.2.7.
- Updated Figure 6 with improved clarity.
- Appendix A.2.1: Added Figure 8 demonstrating how cosine similarity evolves during training.
- Appendix A.2.2: Added new experimental results on evaluating TAM on MobileNet and ViT.
- Appendix A.2.3: Added continual learning experiment comparing TAM and SGDM for training ResNet50 on CLEAR benchmark. We have also updated the conclusion section accordingly.
- Appendix A.2.4: Conducted ablation experiment on varying $\gamma$ to show that it has minimal effect on overall performance.
- Appendix A.2.5: Conducted ablation experiment by defining a variant of AdaTAM that uses exponential moving average to update momentum.
- Appendix A.2.6: Additional gradient norm analysis similar to Figure 6 for comparing TAM and AdaTAM with SGDM and Adam respectively on different datasets.


All the changes made in the revised manuscript are highlighted in blue font.

We also would like to inform that we are also currently running experiments on training GPT. We will add the results in the paper once they are finished.


**<< Post-discussion>>**

We conducted an experiment to compare AdamW and AdaTAMW optimizers in terms of validation loss for training [GPT](https://github.com/karpathy/nanoGPT), using a grid search over three learning rates: {0.006, 0.0006, 0.00006}. The results showed that while both optimizers achieve similar performance by the 100k iters, AdaTAMW converged faster in the early stages.


| Iter  | adamw | adatamw |
|-------|--------------------------|----------------------------|
| 0  | 10.9                     | 10.9                       |
| 1k  | 4.51                     | 4.08                       |
| 5k  | 3.37                     | 3.32                       |
| 10k | 3.20                     | 3.20                       |
| 25k | 3.11                     | 3.11                       |
| 100k | 3.01                     | 3.01                       |

---

### Meta-Review · Area_Chair_BG3M · 2024-12-22

**Metareview:**

This paper proposes TAM, an enhancement to momentum-based optimizers that mitigates oscillations caused by misaligned gradients during training. TAM introduces a damping factor based on the alignment angle between current gradients and previous momentum, stabilizing updates, improving convergence, and exploring the loss landscape better.

Implemented as TAM (for SGDM and Adam(W)), the method is evaluated on tasks like image classification, language model fine-tuning, and online learning. Experiments demonstrate improvements over standard momentum-based methods.

**I appreciate the authors’ efforts during the rebuttal period and the additional experimental results provided**. Given that some reviewers did not respond, I have also reduced the weight of their opinions in my decision-making. However, the paper still has significant issues that require major revisions.

As highlighted by Reviewer fkEA, the proposed algorithm lacks the necessary convergence analysis. Additionally, several reviewers noted the limited experimental scale, overly simple and outdated task settings, and the unclear significance of the performance improvements. The paper also lacks more detailed experimental analyses, such as comparisons of computational overhead and additional reporting of key metrics.

Overall, most reviewers leaned toward rejecting the paper, and the issues raised cannot be resolved through minor revisions. Therefore, I cannot recommend acceptance for the current version.

**Additional Comments On Reviewer Discussion:**

During the rebuttal period, most reviewers initially leaned toward rejecting the paper. Most concerns were about the lack of experiments in modern settings, marginal performance improvements with TAM, insufficient citations and comparisons to recent work, and the absence of theoretical analysis to support the proposed method. While the authors made substantial efforts to address these concerns in their rebuttal, the responses were not sufficient to change the majority opinion. Although some reviewers raised their scores, the overall consensus remained inclined toward rejecting the paper.

---

### Decision · Program_Chairs · 2025-01-22

Reject